# ELLIPTIC LOSS REGULARIZATION

**Ali Hasan**[1,2,*]**, Haoming Yang**[2,*]**, Yuting Ng**[2]**, Vahid Tarokh**[2]
[1] Machine Learning Research, Morgan Stanley
[2] Department of Electrical and Computer Engineering, Duke University
`{ali.hasan, haoming.yang, yuting.ng, vahid.tarokh}@duke.edu,`

## ABSTRACT

Regularizing neural networks is important for anticipating model behavior in regions of the data space that are not well represented. In this work, we propose a regularization technique for enforcing a level of smoothness in the mapping between the data input space and the loss value. We specify the level of regularity by requiring that the loss of the network satisfies an elliptic operator over the data domain. To do this, we modify the usual empirical risk minimization objective such that we instead minimize a new objective that satisfies an elliptic operator over points within the domain. This allows us to use existing theory on elliptic operators to anticipate the behavior of the error for points outside the training set. We propose a tractable computational method that approximates the behavior of the elliptic operator while being computationally efficient. Finally, we analyze the properties of the proposed regularization to understand the performance on common problems of distribution shift and group imbalance. Numerical experiments confirm the promise of the proposed regularization technique.

## 1 INTRODUCTION

Designing effective losses is a longstanding goal for training neural networks. Standard techniques such as Empirical Risk Minimization (ERM) are commonly used for training, but these may overfit to finite samples of data and do not provide any explicit regularization for regions outside the support of the training data. This can be an issue in the case of overparameterized neural networks, which can minimize the loss up to arbitrary accuracy for data points in the training set but have unknown behavior outside the points in the training set. For real-world deployment scenarios, this can be especially concerning where data shifts and imbalances may be present. In response to this limitation, various techniques have been proposed to regularize the loss landscape as a function of input samples of a classifier, e.g. (Hernández-García and König, 2018; Wang et al., 2021; Balestriero et al., 2022).

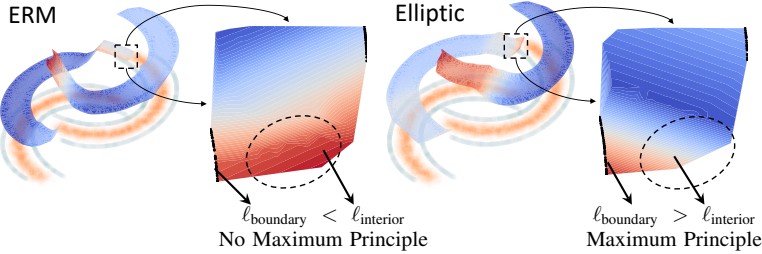

Figure 1: Loss surface of the two moons dataset. The scattered points illustrate the training (blue boundary) and testing (orange interior) samples. The surface represents the loss values of a well-trained classifier that classifies samples to their respective moon. Zooming in on the interior of the ERM loss surface, the training loss exceeds the loss of boundary (circled area) whereas elliptic regularization bounds the loss via the maximum principle of elliptic PDEs.

In this paper, we consider a viewpoint for regularization that stems from the properties of elliptic differential operators where we impose that the loss satisfies an elliptic partial differential equation with boundary data given by the loss. By imposing a loss that satisfies this operator, we can interpret

---

*equal contribution

the expected loss through the lens of the powerful theory developed for elliptic operators and control the parameters of the operator to enforce desirable properties of the learned function approximator.

**Why the PDE Perspective?**    PDEs are useful for characterizing the joint rates in change of different variables. Imposing constraints on how the loss changes as a function of perturbations in the input space is helpful in understanding the behavior of the learned function approximator under different transformations. Additionally, minimizing the loss under the PDE constraints allows us to use the tools from PDEs to qualitatively describe properties of the loss function under the estimated function; an example is given in figure 1 where we show how the *maximum principle* of elliptic operators is used to bound the loss of the estimator. This provides valuable insights into the robustness of the model for regions outside of the training data. To that end, we are mainly interested in the qualitative behavior of the operator and its implications in two major problems related to generalization in machine learning:

> **Data Shifts:** How does the operator influence the loss under *shifts* in the data distribution?
> **Data Imbalance:** How does the operator influence the loss for *underrepresented* groups?

More explicitly, suppose we have data pairings of $(X, y)$ with features $X$ and target variables $y$. Data shifts are data that are collected under different conditions but represent the same phenomena, i.e. $(T(x), y)$ for different transformations $T$ applied to $X$. Data imbalance is the existence of subgroups of $y$ within the dataset that may be more or less frequently represented. By carefully choosing the parameters of the elliptic operator, we can show how behavior of the approximator can be anticipated under these regimes.

To summarize, by grounding our regularization in the rich theory of elliptic PDEs, we provide a principled approach for understanding the generalization properties of neural networks, specifically:

1. We describe a new regularization that corresponds to the solution of an elliptic PDE;
2. We theoretically characterize the practical properties of this regularization through PDE theory;
3. We introduce an efficient computational approach that endows the properties of the elliptic regularization.

## 2    BACKGROUND

Our goal is to investigate a loss that promotes functions that address the two above problems while minimally affecting performance related to our target tasks. We focus on problems of estimating a mapping between features and target variables, such as those in classification and regression tasks. To impose the desired regularity on our mapping, we can consider a few different techniques. On one hand, we can directly restrict the hypothesis class of functions from which we estimate our mapping to ensure those contain the relevant properties. On the other hand, it may be difficult to a priori know which class of functions is sufficient for the approximation task. We instead consider an approach that regularizes the loss over the data space without explicitly constraining the function class, which is implemented through a specific data augmentation.

**Related Work**   The connection between regularization and data augmentation has been studied in a number of settings (Balestriero et al., 2022; Geiping et al., 2022). A particular instance of augmentations is the *mixup* algorithm (Zhang et al., 2017). This algorithm has empirically shown to create more robust classifiers that are better suited to perturbations of the data, provide a level of uncertainty estimation with respect to new samples, and improve performance for classes with few samples (Zhang et al., 2017). These properties lead to many extensions of mixup; one of which is mixupE (Zou et al., 2023), which imposes an explicit directional derivative regularization to improve the mixup algorithm. The class of mixup algorithms has been theoretically analyzed from a regularization perspective (Carratino et al., 2022; Zhang et al., 2020). Various statements have been proved such as in Zhang et al. (2020) which showed that the loss at the interpolated points is an upper bound for $\ell_2$ adversarial training loss. The regularization and generalization effect of mixup led to development of mixup based robust optimization methods such as UMIX (Han et al., 2022), which combines mixup and weighting algorithm to achieve robust training in classification;

and c-mixup (Yao et al., 2022a), which samples based on the distribution of regression label distance for adapting mixup to the robust regression task.

Motivated by the theoretical connection between regularization and data augmentation, we propose a PDE based regularization that can be implemented through a specific type of random data augmentation. This new regularization, which we refer to as *elliptic regularization*, is highly flexible and applies to arbitrary nonnegative loss functions in both regression and classification tasks.

Elliptic regularization focuses on improving optimization under the data shifts and data imbalance scenarios which are widely studied. Algorithms such as DRO (Sagawa et al., 2019) and DORO (Zhai et al., 2021) were introduced to optimize over long-tail distributions and be robust against worst-case shifts in the data distribution. JTT (Liu et al., 2021) and UMIX (Han et al., 2022) are two popular multi-stage methods that weigh data during training based on the earlier results. While most of the above algorithms deal with classification tasks, c-mixup has emerged as a robust optimization method over regression (Yao et al., 2022a). These algorithms, including ours, are group-oblivious where the underlying subpopulation that causes data shifts and imbalance is unknown. We discuss group-informed methods in Appendix F.4.

## 3 PROBLEM SETUP

Let's first define a region which we will consider our domain. Given observations $\{(X, y) \in \mathcal{X} \times \mathcal{Y}\}$ with $\mathcal{X} \subset \mathbb{R}^d$, $\mathcal{Y} \subset \mathbb{R}^k$ we will define $\mathcal{D} = \mathcal{D}_X \times \mathcal{D}_y \supset \mathcal{X} \times \mathcal{Y}$ as a subset of $\mathbb{R}^d \times \mathbb{R}^k$ with nonzero Lebesgue measure. For purposes of analysis we will take $\mathcal{D}$ as the convex hull of $\mathcal{X} \times \mathcal{Y}$ denoted as $\mathcal{C}_{\mathcal{X} \times \mathcal{Y}}$. Additionally, denote the empirical measure of points within a training set $\mathcal{X} \times \mathcal{Y}$ as $\delta_{\mathcal{X} \times \mathcal{Y}} \equiv \frac{1}{|\mathcal{X} \times \mathcal{Y}|} \sum_{i=1}^{|\mathcal{X} \times \mathcal{Y}|} \delta_{(X,y)_i}$, where $|\cdot|$ denotes the cardinality of a set. We want to minimize the risk $\ell : \mathcal{D}_y \times \mathcal{D}_y \to \mathbb{R}_+$ over all the points in the training set for a mapping $f_\theta : \mathcal{D}_X \to \mathcal{D}_y$ with parameters $\theta$. We will disregard any implicit regularization associated with the optimization procedure associated with the minimization problem (i.e. the associated regularization when minimizing such problems using gradient descent). With these in mind, we will consider different flavors of the risk minimization problem and analyze the corresponding regularization from imposing additional terms.

Let us compare two frameworks for minimizing a risk $\ell$, the first performs empirical risk minimization (ERM) and the second minimizes the risk after applying a class of transformations $\mathcal{T}_X, \mathcal{T}_y$ parameterized by $\phi$ which we will refer to as transformed empirical risk minimization (TRM):

$$\min_\theta \mathbb{E}_{\delta_{\mathcal{X} \times \mathcal{Y}}}[\ell(f_\theta(X), y)] \quad \text{(ERM)} \qquad \min_\theta \mathbb{E}_\phi \mathbb{E}_{\delta_{\mathcal{X} \times \mathcal{Y}}}[\ell(f_\theta(\mathcal{T}_X^\phi X), \mathcal{T}_y^\phi y)] \quad \text{(TRM)}$$

From equation ERM we can see that this optimization, which takes place over a discrete set of points, relates to the issues we are trying to address. Since $\delta_{\mathcal{X} \times \mathcal{Y}}$ is a set of measure zero, its unclear what the interpolating behavior of $f$ is for any subset of $\mathcal{X}$ or how $\ell(f_\theta(X), y)$ will behave for any subset of $(X, y) \in \mathcal{D}$. Equation TRM on the other hand may cover more of $\mathcal{D}$ depending on how $\phi$ is chosen.

Instead, we can consider a loss that is defined over all of $\mathcal{D}$ by modifying expectation in equation ERM in a number of different ways. One way is explicitly regularizing the class of functions $f_\theta$ such that the behavior of $\ell(f_\theta(X), y)$ over all $\mathcal{D}$ can be anticipated (for example, making $f_\theta$ Lipschitz would constrain the growth between two points). However, explicitly regularizing $f_\theta$ can be difficult since constraining the function class when $f_\theta$ is a neural network may require defining specific architectures. Alternatively, we can borrow techniques often used in computer vision problems (e.g. (Wang et al., 2021; Yang et al., 2023b)) by applying transformations or augmentations $\mathcal{T}_X^\phi, \mathcal{T}_y^\phi$ with parameters $\phi$ to $X, y$ such that a new objective in equation TRM is minimized for some distribution on $\phi$. This also has the benefit that we now can *sample* transformations rather than directly constrain the class of functions. The question now remains: what class of transformations would be useful to regularize $\ell(f_\theta(X), y)$ over $\mathcal{D}$?

## 4 ELLIPTIC LOSS LANDSCAPES

Our answer to this question involves modifying the ERM problem to construct loss that satisfies the elliptic operator. We introduce all components of the operator in three different parts. To first develop intuition behind the loss, we describe an elliptic operator that has only Laplacian terms that we wish

to constrain. Using this example, we then describe how this operator can be imposed through its stochastic representation given by the Feynman-Kac theorem, providing both a connection to the data augmentation regularization initially described and a technique for implementation. Finally, using this stochastic representation, we describe the full method that includes low order terms through an importance sampling framework.

We define the *loss landscape* as a function $u(X, y) : \mathcal{D} \to \mathbb{R}_+$[1]. Our goal is to prescribe the function $u(X, y)$ with a specified level of regularity over the data space in a way that also imbues $\ell(f_\theta(X), y)$ with regularity and thereby obtain the desired properties listed above. We propose doing this by solving a new minimization problem defined by the following equations:

$$\min_\theta u(X, y), \quad (X, y) \in \mathcal{D}$$

$$0 = \sigma \nabla^2 u(X, y), \quad (X, y) \in \mathcal{D} \tag{1}$$

$$u(X, y) = \ell(f_\theta(X), y), \quad (X, y) \in \partial\mathcal{D} \tag{2}$$

where $\nabla$ is taken with respect to $(X, y)$, $\sigma > 0$ is a coefficient related to the Brownian diffusion which will be later discussed, and $\partial\mathcal{D}$ represents the boundary of the domain. The key aspects of this formulation are: **1.** the constraint in equation 1, corresponding to the elliptic operator, ensures a certain level of regularity within the domain beyond the training data; **2.** the boundary condition in equation 2 connects the loss landscape $u$ to the neural network loss $\ell(f_\theta(X), y)$. The regularity imposed by the PDE in equation 1 describes the behavior for regions of the space away from the observations. Specifically, equation 1 represents the steady state of the heat equation, which diffuses the boundary data from equation 2. The diffusive behavior of this equation defines the regularity of the loss based on the *closest points* in the observation set for regions undefined in the observation set. Additionally, the equation provides a direct connection to an Itô diffusion process which we will discuss in the next section.

**Interpreting the loss landscape** The loss landscape $u$ can be understood as the expected loss under a new data point $(X, y) \in \mathcal{D}$. $u$ satisfies an elliptic PDE with boundary data given by the points in the training set. Since the points in the training set provide all the information we have about the particular phenomena we are representing, the information at these points is diffused according to equation 1.

## 4.1 SOLVING THE REGULARIZED PROBLEM

Computing equation 1 requires solving a PDE over a high dimensional space, which can be computationally infeasible. Fortunately, solutions to PDEs of the type in equation 1 have a Monte Carlo representation which allows scalable computation of solutions. This is formalized through the Feynman-Kac formula, which describes the relationship between expectations of stochastic processes and PDEs (Øksendal, 2003; Pardoux and Raşcanu, 2014). For self-containment, we provide additional descriptions of the Feynman-Kac formula as it relates to the problem we are solving in Appendix D. Using this form, the solution to equation 1 can be written as the following expectation

$$u(x, y) := \min_\theta \mathbb{E}\left[\ell(f_\theta(x_\tau), y_\tau) \mid x_0 = x, y_0 = y\right] \tag{3}$$

where $x_t, y_t$ satisfies $\mathrm{d}(x_t, y_t) = \sigma \, \mathrm{d}W_t$, where $\tau = \inf\{t > 0 \mid (x_t, y_t) \in \mathcal{X} \times \mathcal{Y}\}$, $\sigma > 0$ is the diffusion coefficient, and $W_t$ is standard Brownian motion, and sample path distribution $q$. Since $\delta_{\mathcal{X} \times \mathcal{Y}}$ Lebesgue measure zero, we will almost surely never hit a single point in the set, leading to a $\tau$ being infinite. To circumvent this, we place a ball of size $\varepsilon > 0$ around each point and define a new set $(\mathcal{X} \times \mathcal{Y})_\varepsilon \equiv \{x, y \in \mathcal{D} \mid \exists (x', y') \in \mathcal{X} \times \mathcal{Y} \mid \|(x, y) - (x', y')\| < \varepsilon\}$. The hitting time is then defined under this new set as $\tau^{(\varepsilon)} = \inf\{t > 0 \mid (x_t, y_t) \in (\mathcal{X} \times \mathcal{Y})_\varepsilon \cup \partial\mathcal{C}_{\mathcal{X} \times \mathcal{Y}}\}$, and we compute the expectation with respect to these hitting times. To relate this to equation TRM, sample paths of $(x_t, y_t)$ are a particular type of transformation of the data within the domain.

Using a finite number of samples, we write the empirical expectation as:

$$\min_\theta \frac{1}{N} \sum_{i=1}^{N} \sum_{j=1}^{N} \ell\left(f_\theta(x_{\tau_\varepsilon^{(i)}}^{(i)}), y_{\tau_\varepsilon^{(i)}}^{(i)}\right) \quad s.t. \quad x_0^{(i)}, y_0^{(i)} = x^{(j)}, y^{(j)}.$$

---

[1]Note that this is different from the loss landscape definition which is given as a function of training iterations and considers perturbations in the parameter space.

where $\tau_\varepsilon^{(i)} = \inf\{t > 0 \mid (x_t, y_t)^{(i)} \in (\mathcal{X} \times \mathcal{Y})_\varepsilon\}$. Sample paths of $(x_t, y_t)$ are generated using a standard Euler-Maruyama method. Qualitatively, as $\varepsilon$ increases, the behavior of the solution becomes piecewise constant at each point in the training set. As $\varepsilon$ decreases, the solution becomes more diffusive where each data point becomes a point source.

## 4.2 GENERALIZATIONS OF THE OPERATOR

The form of the PDE described in equation 1 is motivated by the diffusive properties of the second order derivative, which has the property that the loss values on the boundaries diffuse at a rate proportional to $\sigma$. Including only second order derivatives is not strictly necessary, however. The equation can be easily generalized to include lower order terms by including an *advection* term which acts as a deterministic term in the sense that information is being propagated according to an expected trajectory rather than diffused. The connection between the stochastic process $(x_t, y_t)$ and the operator easily extends to this case. Through Girsanov's theorem, we can define the Radon-Nikodym derivative $\frac{dp}{dq}$ between two path measures $p \ll q$ to redefine the optimization criterion as

$$u_p(x, y) = \min_\theta \mathbb{E}_q \left[ \ell(f_\theta(x_\tau), y_\tau) \frac{dp}{dq} \mid x_0 = x, y_0 = y \right] \tag{4}$$

which has the effect of reweighting some of the sample paths of $(x_t, y_t)$. This results in $u$ satisfying a new PDE with lower order terms given by the structure of $\frac{dp}{dq}$ in

$$0 = -\frac{1}{2} \sum_{i=1}^{d+k} a_i \frac{\partial^2 u}{\partial z_i^2} + b^\top \nabla u \tag{5}$$

which we assume is parameterized by a drift function $b : \mathcal{D} \to \mathbb{R}^{d+k}$. This reweighting term can then be interpreted with respect to the problems we are trying to solve. For example, regarding the problem of data imbalance, sample paths corresponding to the underrepresented data groups can be reweighed to greater influence the loss. This provides a connection to existing loss functions, such as the focal loss in Lin et al. (2017), where points within the training distribution are weighted according to their loss values if we choose $b$ to be a function of $\ell(f_\theta(X), y)$.

## 5 PRACTICAL CONSIDERATIONS OF THE REGULARIZER

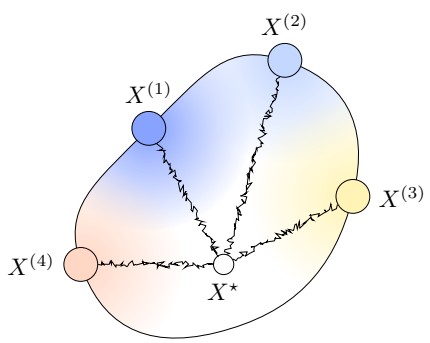

There are a few important practical notes regarding applying elliptic regularization. We will first describe how to efficiently implement this loss and discuss a choice for the importance factor in equation 4 while maintaining the desired properties of the operator. We then theoretically illustrate the main properties of interest and how they are relevant to the problems presented above: The maximum principle which allows us to bound the error on the interior of the domain in Proposition 1; and, the qualitative behavior on the on the loss landscape in regression in cases of distribution shifts and class imbalance in Propositions 2 and 3.

Figure 2: Illustration of the loss values over a domain with 4 points on the boundary. The expected loss at point $X^\star$ is composed of losses at $\varepsilon-$balls around $X^{(i)}, i = 1 \ldots 4$. Black paths represent sample paths starting at $X^\star$.

### 5.1 NUMERICAL PROCEDURE

In practice, it may be difficult to compute the first hitting time of the boundary within a reasonable amount of time. For example, computing the paths up to the first hitting time requires sampling for an undefined amount of time. For a practical implementation, we consider a sampling method that approximates the behavior of the first hitting time but does not require sampling unconstrained sample paths. We require that the approximation **a)** runs in a finite amount of time and **b)** maintains the important properties of the PDE. To do this, we first compute the pairwise distances between all points within a batch according

to some distance metric. Then, for each starting point $(x, y)$, we sample an endpoint according to a discrete distribution $P$ over endpoints with mass inversely proportional to the distances computed. Equation 3 would ordinarily be solved over all $(X, y) \in \mathcal{D}$ (e.g. by sampling uniformly over the space). We instead consider sampling over paths that connect data points using the minimum distance. Figure 2 illustrates an example of this where four points on the boundary are used to define the loss at $X^\star$. The loss is minimized at all points along the paths between the source point $X^\star$ and the target points $X^{(i)}, i = 1 \dots 4$. This is done by sampling Brownian bridges between the pairings and optimize the following loss:

$$\min_\theta \mathbb{E}_{(X_\rightarrow, y_\rightarrow) \times (X_\leftarrow, y_\leftarrow) \sim P(\Pi)} \mathbb{E}_{X_s, y_s \sim \mathrm{BB}_{X_\rightarrow, y_\rightarrow}^{X_\leftarrow, y_\leftarrow}} \left[ \int_0^1 \ell(f_\theta(X_s), y_s) \mathrm{d}s \right] \tag{6}$$

for all $s$, where we denote $\mathrm{BB}_{X_\rightarrow, y_\rightarrow}^{X_\leftarrow, y_\leftarrow}$ as a Brownian bridge sample path where $\rightarrow$ denotes starting points and $\leftarrow$ denotes end points and $\Pi$ is the set of points in the support of $\delta_{(\mathcal{X} \times \mathcal{Y})} \times \delta_{(\mathcal{X} \times \mathcal{Y})}$. This expectation involves solving the problem equation 3 for all points along the Brownian bridge between $X_\rightarrow, y_\rightarrow$ and $X_\leftarrow, y_\leftarrow$. This minimizes the loss over all paths between the data points where the paths satisfy the distribution corresponding to the PDE between the endpoints. This loss, through an application of Dynkin's formula, corresponds to exactly solving equation 3 for points along the Brownian bridge paths, which we formally describe in Lemma 1 in Appendix B. Since the support of the bridges includes the domain, all points within the domain should eventually be sampled. Additionally, the starting point of the bridge can be an arbitrary point within the domain; we use the data points for convenience, but the theoretical properties still hold with this sampling technique. The connection to the transformation based regularization described in equation TRM is made more explicit – $\mathcal{T}$ in equation TRM can be seen as the Brownian bridge transformation on the data points. Note that there exist some numerical error with this approximation, since it requires discretizing a continuous process, we refer to Graham and Talay (2013, Chapter 7.2) for additional details. We describe additional implementations of the bridge regularizer and how they relate to the original elliptic operator in Appendix E.

## 5.2 Bounding the Loss

One property we would like to guarantee on our loss is, for any new point within $\mathcal{D}$, what should we expect the loss to be? For example, under distribution shifts we may want to predict what the expected loss is. We do this through characterizing the loss landscape $u$ as a function of the observed data. Since we impose that $u$ must satisfy a PDE, we can use the properties of the solution of the PDE to bound the solution in the interior of the domain. In particular, since the PDE is elliptic, the maximum principle is satisfied. This allows us to bound the interior of the domain by the values on the boundary (which are our training points). We formalize this in the following proposition:

**Proposition 1** (Bounding the Loss for an Interior Point). *Consider any point $X, y \in \mathcal{D}$ and suppose the function pairs $u, f_\theta$ solves equation 1. Then, the expected loss $u$ at $X, y$ satisfies the following inequality:*

$$\min_{X, y \in \mathcal{X} \times \mathcal{Y}} \ell(f_\theta(X), y) \leq u(X, y) \leq \max_{X, y \in \mathcal{X} \times \mathcal{Y}} \ell(f_\theta(X), y).$$

*Proof.* The proof follows from the fact that $u$ satisfies an elliptic PDE that satisfies both the maximum and minimum principle. Therefore, all extreme points lie on the boundary of the domain which correspond to the points in the training set. $\qquad\square$

This has an important implication insofar as we can anticipate the behavior of our loss landscape $u$ for new, unseen points by guaranteeing the loss is bounded by the training set points. This condition holds for any point within the domain $\mathcal{D}$.

## 5.3 Effects on Distribution Shifts and Data Imbalance

Finally, we discuss how to interpret the elliptic operator for the problems of interest involving distribution shift and class imbalance which motivate this work. We focus on describing the behavior for the regression case where the loss is given by the $\ell^2$ distance and the function $f$ is a two-layer ReLU neural network and study the loss landscape $u(X, y)$ satisfying the following conditions:

$$\sigma \nabla^2 u(X, y) = 0, \qquad\qquad X, y \in \mathcal{D} \tag{7}$$
$$u(X, y) = (f(X) - y)^2, \qquad\qquad X, y \in \partial\mathcal{D}$$

with $f(X) = W_1 \mathrm{ReLU}(W_0 X)$ and where $W_0 \in \mathbb{R}^{K \times d}$ and $W_1 \in \mathbb{R}^{K \times 1}$.

**Applied to data shifts** Suppose we define a set of transformations $T_\varphi : \mathcal{X} \to \mathcal{X}$ parameterized by $\varphi \subset \Phi$ for some space of parameters $\Phi$ and require that these transformations only affect the input features $X$ but do not affect the target variable $y$. As examples, we can think of medical imaging data collected at different hospitals under different operating conditions but of the same disease and of the same modality with the target variable being the class of disease in the image. Certain parameter values of $\varphi$ may be sampled more in the dataset than others (e.g. data from larger hospitals) leading to greater uncertainty on samples from the underrepresented parameter values. We will denote an example subset with few sample points $\Phi_\wedge$. As such, the subset of $\mathcal{X}$ associated with $T_\varphi(X), \varphi \in \Phi_\wedge$ will have small cardinality. Then, the expected hitting time of a point within the space is longer since fewer points are within the vicinity of a point within this subset.

In the following proposition, we formalize this idea and study the changes in the loss landscape when an affine transformation is applied to the features:

**Proposition 2** (Expected Error Under Affine Transformations). *Consider the loss landscape $u$ satisfying equation 7 and consider the class of affine transformations $\mathcal{T}$ where $T(X) = A_T X + b_T \in \mathcal{T}$ for $A_T \in \mathbb{R}^{d \times d}$ and $b_T \in \mathbb{R}^d$ and $(X, y) \in \mathcal{D} \neq \mathcal{X} \times \mathcal{Y}$. Suppose the true error is given by $(f(X) - y)^2$ and $f$ is $C$-Lipschitz. Then, the loss landscape $u$ satisfies*

$$u(T(X), y) \leq 2 W_1 W_0 (W_1 W_0)^\top \tau_T + C|\Delta|(|\Delta| + 2\varepsilon),$$

*where $\varepsilon := |f(X) - y|$, $\Delta := A_T X + b_T - X$, and $\tau_T = \inf_{t>0}\{X_t, y_t \in \partial \mathcal{D} \mid X_0 = A_T X + b_T, y_0 = y\}$.*

Proposition 2 allows us to anticipate the behavior of our loss under affine distribution shifts of our data. By applying this regularization, we can guarantee that the expected loss is no greater than a function of the parameters of the neural network used for regression.

**Applied to data imbalance** With regards to imbalanced data in the classification setting, we consider a local structure on the points within each class. We will partition $\mathcal{Y}$ into two classes, $\mathcal{Y}_\vee$ and $\mathcal{Y}_\wedge$ for the over-represented and under-represented classes respectively. We can show that when the loss landscape satisfying the operator in equation 7, the expected loss is greater for regions of low data density versus for regions of high data density.

Following a similar argument as in Proposition 2, we note that the hitting times for $\mathcal{Y}_\wedge$ is greater than the ones in $\mathcal{Y}_\vee$ with certain probability. The proposition is based on a probabilistic interpretation of how many components of the weight matrices in the neural network are greater than or equal to zero.

**Proposition 3** (Expected Error in Regions of Low Density). *Consider the loss landscape $u(X, y)$ satisfying equation 7. Let $q(\epsilon) := P(\nabla^2 (f(X) - y)^2 \geq 2\epsilon W_1 W_0 (W_1 W_0)^\top)$ be a continuous function, then an $\epsilon^\star \in (0, 1)$ such that $q(\epsilon^\star) = 1 - \epsilon^\star$ exists. Choose the smallest $\epsilon^\star$. Let $(X_\vee, y_\vee)$ and $(X_\wedge, y_\wedge)$ be two points where $y_\wedge$ is in a class that is well represented such that for $\tau_{y_\vee} = \inf_{t>0}\{(X_t, y_t) \in \mathcal{D} \mid X_0 = X_\vee, y_0 = y_\vee\}$ and $\tau_{y_\wedge} = \inf_{t>0}\{(X_t, y_t) \in \mathcal{D} \mid X_0 = X_\wedge, y_0 = y_\wedge\}$ the relation $\tau_\wedge \geq \frac{\tau_{y_\vee}}{\epsilon^\star}$ holds. Suppose also that $(f(X_\vee) - y_\vee)^2 = (f(X_\wedge) - y_\wedge)^2$. Then, $u(X_\vee, y_\vee) \leq u(X_\wedge, y_\wedge)$ with probability $1 - \epsilon^\star$.*

Proposition 3 states that within regions of low data density, the loss landscape to be greater than in regions of high density, which is important in cases involving data imbalance.

## 6 EXPERIMENTS

We present empirical results over a comprehensive set of tasks related to the proposed regularization scheme. We first empirically evaluate the bound derived from Proposition 1 and verify that the proposed regularization retains the necessary loss within the domain of interest. Next, we benchmark the elliptic regularization against another popular regularization scheme known as mixup (Zhang et al., 2017) and its variants on balanced classification and in-distribution regression. We then investigate the benefits of elliptic training on classification and regression with imbalanced group populations, imbalanced domains, and subdomain shifts. Finally, we experiment with elliptic training on imbalanced classification. Additional comparisons and ablation studies are provided in

Table 1: Classification error on CIFAR-10, CIFAR-100, best method is **bolded** while the second best is underlined

|  |  | CIFAR-10 | CIFAR-100 |
|---|---|---|---|
| PreActRN50 | ERM | $4.71_{\pm0.06}$ | $24.68_{\pm0.4}$ |
|  | mixup | $4.53_{\pm0.04}$ | $23.03_{\pm0.5}$ |
|  | mixupE | $\mathbf{3.53}_{\pm0.05}$ | $\underline{20.23}_{\pm0.5}$ |
|  | Elliptic | $\underline{4.15}_{\pm0.43}$ | $\mathbf{18.95}_{\pm0.3}$ |
| PreActRN101 | ERM | $4.21_{\pm0.07}$ | $23.20_{\pm0.4}$ |
|  | mixup | $4.43_{\pm0.05}$ | $23.05_{\pm0.4}$ |
|  | mixupE | $\mathbf{3.35}_{\pm0.05}$ | $\mathbf{18.86}_{\pm0.4}$ |
|  | Elliptic | $\underline{3.56}_{\pm0.2}$ | $\underline{19.03}_{\pm0.6}$ |
| Wide-RN28 | ERM | $4.24_{\pm0.1}$ | $22.20_{\pm0.1}$ |
|  | mixup | $\underline{3.03}_{\pm0.09}$ | $19.38_{\pm0.1}$ |
|  | mixupE | $\mathbf{2.94}_{\pm0.05}$ | $\mathbf{17.12}_{\pm0.1}$ |
|  | Elliptic | $3.09_{\pm0.1}$ | $\underline{17.71}_{\pm0.1}$ |

Table 2: Classification accuracy on Tiny-Imagenet 200, best method is **bolded** while the second best is underlined

|  |  | Top-1 (%) | Top-5(%) |
|---|---|---|---|
| PreActRN18 | ERM | $54.97_{\pm0.5}$ | $72.71_{\pm0.5}$ |
|  | mixup | $54.65_{\pm0.4}$ | $72.53_{\pm0.5}$ |
|  | mixupE | $\underline{62.21}_{\pm0.4}$ | $\underline{82.09}_{\pm0.4}$ |
|  | Elliptic | $\mathbf{65.03}_{\pm0.1}$ | $\mathbf{84.98}_{\pm0.1}$ |
| PreActRN34 | ERM | $57.25_{\pm0.5}$ | $72.58_{\pm0.5}$ |
|  | mixup | $57.79_{\pm0.4}$ | $76.15_{\pm0.4}$ |
|  | mixupE | $\underline{65.37}_{\pm0.3}$ | $\underline{83.77}_{\pm0.4}$ |
|  | Elliptic | $\mathbf{66.67}_{\pm1.7}$ | $\mathbf{85.71}_{\pm1.2}$ |
| PreActRN50 | ERM | $55.91_{\pm0.6}$ | $73.50_{\pm0.6}$ |
|  | mixup | $54.86_{\pm0.5}$ | $73.11_{\pm0.4}$ |
|  | mixupE | $\underline{67.22}_{\pm0.4}$ | $\underline{85.14}_{\pm0.4}$ |
|  | Elliptic | $\mathbf{70.28}_{\pm0.1}$ | $\mathbf{88.14}_{\pm0.2}$ |

Appendix F with detailed data descriptions for all datasets in Appendix G. All hyperparameter settings are described in Appendix H. We note that although the implementation described in Section 5.1 specifies computing pairwise distances between data, we empirically show in Appendix F that this distance computation has a minor effect on performance of the method.

## 6.1 EMPIRICAL STUDY OF PROPOSITION 1

Using the two moons dataset as an example, we empirically illustrate that optimizing the neural network by solving the PDE through equation 6 enforces the bound in Proposition 1.

We first train a 2-layer, 4-hidden unit multilayer perceptron using the different optimization criteria of ERM, elliptic, and mixup with the generated boundary data (blue in Figure 3). We then compute the loss on the within-boundary data (orange in Figure 3). We sampled 10000 points on the boundary to train ERM and mixup while using 1000 boundary data with 10 discretized timesteps for the elliptic loss[2] to sample Brownian bridges. We show in Figure 3 that only the neural network trained using equation 6 maintains the interior loss within the bounds of the boundary loss as specified in Proposition 1.

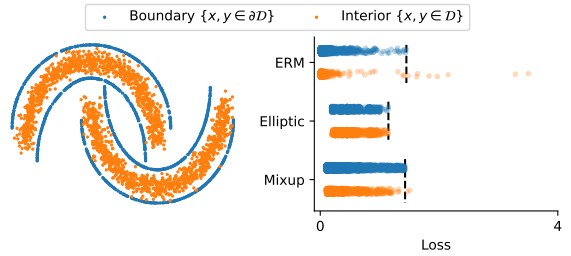

Figure 3: Comparison of loss for data on boundary vs within the boundary of the data space. The dashed line indicates the max loss of the boundary data.

## 6.2 EXPERIMENTS ON REGRESSION AND CLASSIFICATION

We now consider how the proposed elliptic regularization performs on standard regression and classification tasks in machine learning. While we are interested in understanding the regimes in which the regularization scheme improves the performance for the cases of group imbalance and distribution shift, we are also interested in seeing how the performance is in "normal" regimes without these issues. To do this, we evaluate elliptic training against suitable baselines of vanilla mixup (Zhang et al., 2017) and related state-of-the-art mixup algorithms for classification (Zou et al., 2023) and regression (Yao et al., 2022a).

We then consider the performance when applying the Brownian bridge algorithm described in Section 5.1 without additional importance weighting. We showcase that using the elliptic regularization achieves comparable results to the state-of-the-art mixup algorithms for these balanced, in-distribution experiments. We benchmark on CIFAR-10 and CIFAR-100 with classification error rate in Table 1;

---

[2]Two endpoints and 8 diffusion steps.

we also evaluate on tiny-Imagenet 200 and report the top-1 and top-5 accuracy in Table 2. A number of different architectures are used in this experiment including PreActResNet of various depths (He et al., 2016) and Wide-Resnet-28-10 (Zagoruyko and Komodakis, 2016). The experiment was repeated with 5 random seeds. The results suggest that elliptic training generally improves performance over the vanilla mixup algorithm and improves upon mixupE on the more complicated Tiny-Imagenet 200. The results also suggest that the elliptic regularization may be most helpful in datasets with a larger number of classes. This may be because the interpolation between classes will more likely lie on the interior of the simplex rather than on discrete corners, as is the case for datasets with fewer classes.

For regression, we benchmarked the elliptic regularization against c-mixup algorithm on two real world tabular datasets: Airfoil Self-Noise (Airfoil) (Brooks et al., 2014) and NO2 (Kooperberg, 1997); Airfoil contains aerodynamic features of airfoil blade and related acoustic statistics and NO2 studies the concentration of NO2 particles given traffic volume and meteorological variables. Average RMSE is reported in Table 3, and the standard deviations are calculated over 10 repetitions.

Table 3: Average Regression (RMSE) on Regression

| Algorithm | AirFoil | NO2 |
|---|---|---|
| C-mixup (Yao et al., 2022a) | $2.88_{\pm 0.187}$ | $0.524_{\pm 0.006}$ |
| Elliptic | $\mathbf{2.85}_{\pm 0.174}$ | $\mathbf{0.523}_{\pm 0.008}$ |

### 6.3 Robustness to Group Imbalance and Distribution Shift

We now explicitly apply the elliptic regularization to the loss function to evaluate the performance on data with imbalanced subpopulations and under distribution shifts. We continue to evaluate on both classification and regression tasks. For the robust classification task, we benchmark against two types of algorithms: 1) single-stage training algorithm including Focal Loss (Lin et al., 2017), CVaR-DRO and $\chi^2$-DRO (Levy et al., 2020), CVaR-DORO and $\chi^2$-DORO (Zhai et al., 2021); and 2) two-stage training algorithm including JTT (Liu et al., 2021) and UMIX (Han et al., 2022). These are group-oblivious algorithms that train without subpopulation/domain information; a full comparison with group-informed algorithms on classification tasks is provided in appendix F.4. We benchmark elliptic training with importance weighting drift $b = \nabla \ell(f(x), y)$ in equation 5 on WaterBirds (Koh et al., 2021) and CelebA (Sagawa et al., 2019) to evaluate performance on robustness for group imbalance. We also evaluate on the Camelyon17 (Bandi et al., 2018) dataset to examine robustness under domain shifts. To remain consistent with previous literature, we report average and worst group accuracy for Waterbirds and CelebA across 3 random seeds and report 10 seed average accuracy for Camelyon17. The classification results are presented in Table 4. The elliptic regularization greatly improves upon the one-stage algorithms and is comparable to the two-stage algorithm Just-Train-Twice (Liu et al., 2021), which requires two separate training sessions of the model.

Table 4: Robust classification accuracy under imbalance subpopulations and domain shift; best methods are **bolded** and best one-stage methods are underlined.

| Algorithm | WaterBirds | | CelebA | | Camelyon17 |
|---|---|---|---|---|---|
| | Avg(%) | Worst(%) | Avg(%) | Worst(%) | Avg(%) |
| Focal Loss (Lin et al., 2017) | $87.0_{\pm 0.5}$ | $73.1_{\pm 1.0}$ | $88.4_{\pm 0.3}$ | $72.1_{\pm 3.8}$ | $68.1_{\pm 4.8}$ |
| CVaR-DRO (Levy et al., 2020) | $90.3_{\pm 1.2}$ | $77.2_{\pm 2.2}$ | $86.8_{\pm 0.7}$ | $76.9_{\pm 3.1}$ | $70.5_{\pm 5.1}$ |
| CVaR-DORO (Zhai et al., 2021) | $91.5_{\pm 0.7}$ | $77.0_{\pm 2.8}$ | $89.6_{\pm 0.4}$ | $75.6_{\pm 4.2}$ | $67.3_{\pm 7.2}$ |
| $\chi^2$-DRO (Levy et al., 2020) | $88.3_{\pm 1.5}$ | $74.0_{\pm 1.8}$ | $87.7_{\pm 0.3}$ | $\underline{78.4}_{\pm 3.4}$ | $68.0_{\pm 6.7}$ |
| $\chi^2$-DORO (Zhai et al., 2021) | $89.5_{\pm 1.0}$ | $76.0_{\pm 3.1}$ | $87.0_{\pm 0.6}$ | $75.6_{\pm 3.4}$ | $68.0_{\pm 7.5}$ |
| Elliptic + IW | $\underline{92.0}_{\pm 0.3}$ | $\underline{84.1}_{\pm 1.1}$ | $\mathbf{91.3}_{\pm 0.3}$ | $77.4_{\pm 4.5}$ | $\mathbf{\underline{77.9}}_{\pm 3.0}$ |
| Two-stage: JTT (Liu et al., 2021) | $\mathbf{93.6}_{\pm NA}$ | $86.0_{\pm NA}$ | $88.0_{\pm NA}$ | $81.1_{\pm NA}$ | $69.1_{\pm 6.4}$ |
| Two-stage: UMIX (Han et al., 2022) | $93.0_{\pm 0.5}$ | $\mathbf{90.0}_{\pm 1.1}$ | $90.1_{\pm 0.4}$ | $\mathbf{85.3}_{\pm 4.1}$ | $75.1_{\pm 5.9}$ |

For the regression task, we compare against the c-mixup (Yao et al., 2022a) algorithm. We experiment on SkillCraft and Crime to examine domain shift and RCF-MNIST for sub-domain shift. Communities and Crime (Crime) (Redmond, 2009) and SkillCraft1 Master Table (SkillCraft) Blair et al. (2013) are real-world tabular datasets where domain shifts exist between the training and testing data. Detailed description of these dataset is provided in Appendix G. All experiments for regressions are repeated with 10 different random seeds; we report the average and worst RMSE in Table 5.

Table 5: Regression (RMSE) performance under (sub)domain shifts datasets; best method is **bolded**.

| Algorithm | SkillCraft | | Crime | | RCF-MNIST | |
|---|---|---|---|---|---|---|
| | Avg | Worst | Avg | Worst | Avg | Worst |
| C-mixup (Yao et al., 2022a) | $6.27_{\pm 0.537}$ | $\mathbf{8.83}_{\pm 1.010}$ | $0.132_{\pm 0.003}$ | $0.167_{\pm 0.010}$ | $0.165_{\pm 0.001}$ | $0.180_{\pm 0.001}$ |
| Elliptic | $\mathbf{5.97}_{\pm 0.283}$ | $9.17_{\pm 1.150}$ | $0.132_{\pm 0.003}$ | $\mathbf{0.164}_{\pm 0.01}$ | $\mathbf{0.162}_{\pm 0.002}$ | $\mathbf{0.178}_{\pm 0.002}$ |

## 6.4 CLASS IMBALANCE AND ROBUSTNESS TO NOISE

We further evaluate the efficacy of elliptic regularization from two perspectives: imbalance classification (average and worst class) and robustness to noise. We focus on the real-world medical dataset Med-MNIST (Yang et al., 2023a) where class imbalance exists due to the prevalence of different diseases in the patient populations. We chose 4 sub-datasets of the Med-MNIST dataset to evaluate the effectiveness of the elliptic regularization. Detailed descriptions of these datasets are provided in Appendix G. To showcase the robustness induced by elliptic training, we include 50% label noise to the training data where half of the training data labels are randomly shuffled. Average and worst-class accuracy over 10 seeds are presented in Table 6, where we note the significant improvement of elliptic training and further improvement brought forth by importance weighting. The results in Table 6 indicate that elliptic training performs well in the class-imbalance tasks while still maintaining high performance under large label noise.

Table 6: Average and worst class classification accuracy for selected Med-MNIST dataset, best method is **bolded** while the second best is underlined

| | Breast | | Blood | | Path | | OrganC | |
|---|---|---|---|---|---|---|---|---|
| Method | Avg(%) | Worst(%) | Avg(%) | Worst(%) | Avg(%) | Worst(%) | Avg(%) | Worst(%) |
| ERM | $80.0_{\pm 3.4}$ | $32.4_{\pm 13.8}$ | $81.4_{\pm 2.7}$ | $56.3_{\pm 17.3}$ | $55.7_{\pm 3.8}$ | $10.7_{\pm 8.6}$ | $84.1_{\pm 1.9}$ | $65.1_{\pm 8.2}$ |
| mixup | $83.1_{\pm 2.4}$ | $47.1_{\pm 10.9}$ | $78.8_{\pm 3.1}$ | $41.7_{\pm 19.3}$ | $54.3_{\pm 2.1}$ | $1.2_{\pm 2.1}$ | $85.6_{\pm 2.4}$ | $56.8_{\pm 11.0}$ |
| mixupE | $76.5_{\pm 2.8}$ | $16.7_{\pm 11.3}$ | $74.2_{\pm 3.6}$ | $31.9_{\pm 16.9}$ | $52.5_{\pm 2.0}$ | $2.7_{\pm 7.6}$ | $70.4_{\pm 7.6}$ | $35.6_{\pm 16.8}$ |
| Elliptic | $\mathbf{87.6}_{\pm 1.8}$ | $\underline{67.6}_{\pm 4.4}$ | $\mathbf{85.5}_{\pm 1.6}$ | $\underline{60.2}_{\pm 14.9}$ | $\underline{62.3}_{\pm 2.5}$ | $\mathbf{26.4}_{\pm 10.2}$ | $\underline{87.7}_{\pm 0.9}$ | $\underline{65.9}_{\pm 3.4}$ |
| + IW | $\underline{87.3}_{\pm 0.9}$ | $\mathbf{67.6}_{\pm 3.0}$ | $\underline{84.1}_{\pm 4.0}$ | $\mathbf{68.0}_{\pm 13.7}$ | $\mathbf{62.7}_{\pm 1.8}$ | $\underline{20.9}_{\pm 6.8}$ | $\mathbf{88.0}_{\pm 0.7}$ | $\mathbf{66.2}_{\pm 4.8}$ |

## 7 DISCUSSION

In this work, we proposed a regularization technique for the loss landscape over a defined domain where we borrow ideas from PDEs to impose properties on the landscape. We require that the loss landscape satisfies an elliptic operator and described computational tools for enforcing this. The operator allows us to bound the expected loss on the interior of the domain using the points that we observe while also providing theoretical behavior for the loss in distribution shifts and class imbalance. There exist many avenues for extending the work provided. A further study on the properties of Radon-Nikodym derivative corresponding to the parameters of the first order derivatives of the PDE should be undertaken. We chose the current parameters due to their ease in computation, but there may be more ways one can impose coefficients. Related to this, an important theoretical investigation involves studying the coefficients of the operator from a stochastic control perspective to understand the behavior under, for example, stochastic gradient descent. One can easily show that the gradient with respect to parameters satisfies another elliptic PDE over the space of inputs using similar tools of analysis, but the implications on the training behavior of the parameters is still not clear. It could be helpful to link to other theories, e.g. flatness of minima, to understand the effects on the parameter space; or other types of PDEs, such as the parabolic PDEs, to provide another avenue of study on other learning regimes (Yang et al., 2025). Finally, extensions to data supported on manifolds could provide an interesting avenue for regularization for data on complex geometries.

**Limitations** There are a few limitations with the proposed framework. Without using the Brownian bridge technique, solving the PDE may have infinite time until hitting the first point on the boundary. It is therefore advisable to study what happens when including boundary conditions (e.g. reflecting boundaries) or imposing drift such that the diffusion is guaranteed to hit a point on the boundary. Additionally, the computation time is greater for sampling these stochastic processes than with ERM.

ACKNOWLEDGEMENT

Ali Hasan and Vahid Tarokh were supported in part by the Air Force Office of Scientific Research (AFOSR) under award number FA9550- 20-1-0397. Haoming Yang was supported in part by Air Force Office of Scientific Research (AFOSR) under award number FA9550-22-1-0315

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

ELLIPTIC OPERATORS IN LOSS LANDSCAPES (SUPPLEMENTARY MATERIAL)

## A  ALGORITHM DETAILS

To fully supplement the algorithmic contributions in the main paper, we detail the elliptic training procedure that solves the PDE of equation 1 using Brownian bridges in Algorithm 1. There are several hyperparameters for the training algorithm, such as the number of bridges $n_b$, the number of time discretization steps $n_t$, the bridge diffusion constant $\sigma_b$, and the strength of importance weighting $\xi$. We will present some ablation studies exploring the effect of hyperparameter $\xi$ and $\sigma_b$ in Section F.1. We also note that depending on different tasks, there could be constraints imposed on the Brownian bridges of $y$. For example, for a classification task, $y \in [0, 1]$, so we project the Brownian bridge of $y$ onto the simplex by taking the absolute value of the path and normalizing it.

---

**Algorithm 1** Elliptic training algorithm

---

Input: Data $X_{\text{all}}, y_{\text{all}}$, related Brownian bridge hyperparameters.
Initialize: neural network $f_\theta$.
Compute pairwise $l_2$-distance between all data in $X_{\text{all}}$ (optional).
**for** $X, y$ in mini-batched $X_{\text{all}}, y_{\text{all}}$ **do**
    Obtain Brownian bridge pairs $X', y'$ by sampling inversely proportional to distance computed
    Sample Brownian bridges $X_s, y_s \sim \text{BB}_{X,y}^{X',y'}$ for arbitrary timestep $s$ (see Algorithm 2)
    Compute loss $\ell$ as equation 6 with $\ell(f(X_s), y_s)$ using the Euler's method over $s \in [0, 1]$.
    **if** use importance weight **then**
        Compute gradient $\nabla \ell$, then compute $\ell = \ell + \xi \nabla \ell$
    **end if**
    Minimize $\ell$ using gradient-based optimizer
**end for**

---

To sample a brownian bridge, we used the Euler-Maruyama Scheme below

---

**Algorithm 2** Sampling a Brownian bridge

---

**Input:** Initial condition $X \in \mathbb{R}^d$, terminal condition $X' \in \mathbb{R}^d$, diffusion coefficient $\sigma$, number of time steps $M$
Let $\Delta_t = \frac{T - t_0}{M}$ where $t_0 = 0, t_M = 1$, and $t_{i+1} - t_i = \Delta_t$
Sample $M$ samples from standard normal $N(0, I)$, denoted as $\Delta W_t$
Integrate samples to form Brownian motion $W_t = \sum_{s=0}^{t} \sigma \Delta W_s \sqrt{\Delta_t}$
Set $W_0 = 0$; then compute $W_t^X = W_t + X$
**Output:** Brownian bridge as $\text{BB}_t = W_t^X - t(W_M^X - X')$

---

### A.1  COMPUTATION COMPLEXITY

The most computationally expensive part of our method is computing pairwise distance for every dataset, which is of time-complexity $\mathcal{O}(n^2)$. However, this can be pre-computed and saved as a dictionary to be called for $\mathcal{O}(1)$ speed during training. We also show, through empirical experiment, the computation of distance may not be necessary.

We also tested the computation time between UMIX and the proposed method on the waterbirds dataset. All hyperparameters of UMIX are taken from Han et al. (2022). Three run average training time with the proposed method is 8345 seconds while UMIX requires 11795 seconds (stage 1: 5918, stage 2: 5877). This result suggests that the additional computational cost associated with sampling bridges is marginal and the proposed elliptic regularization remains competitive in performance with a lower training time.

## B  PROOFS

### B.1  PROOF OF PROPOSITION 2

*Proof.* We will assume $\tau < \tau_T$, where $\tau_T$ is the stopping time for the Brownian motion when starting at the transformed coordinates. This is a natural assumption since the transformed data may be further from other points after transformation and result in a longer hitting time. From Dynkin's formula, we can write the solution in terms of the stopping time and the boundary condition as:

$$\mathbb{E}[(f(X) - y) \mid X_0 = T(X), y_0 = y] = (f(T(X)) - y)^2$$
$$+ \mathbb{E}\left[\int_0^{\tau_T} \nabla^2 (f(X_s) - y)^2 \mathrm{d}s \mid X_0 = T(X), y_0 = y\right].$$

Since $\nabla^2 (f(X_s) - y)^2 \leq 2W_1 W_0 (W_1 W_0)^\top$, we rewrite the solution as

$$\mathbb{E}[(f(X) - y) \mid X_0 = T(X), y_0 = y] \leq (f(T(X)) - y)^2 + 2W_1 W_0 (W_1 W_0)^\top \tau_T.$$

Compared with the original expected error given by

$$(f(X) - y)^2 \leq \mathbb{E}[(f(X) - y) \mid X_0 = X, y_0 = y] \leq (f(X) - y)^2 + 2W_1 W_0 (W_1 W_0)^\top \tau.$$

Taking the difference, we get

$$
\begin{aligned}
error &\leq 2W_1 W_0 (W_1 W_0)^\top \tau_T + (f(T(X)) - y)^2 - (f(X) - y)^2 \\
&\leq 2W_1 W_0 (W_1 W_0)^\top \tau_T + (f(T(X)) - f(X))(f(T(X)) + f(X) - 2y) \\
&\leq 2W_1 W_0 (W_1 W_0)^\top \tau_T + |f(T(X)) - f(X)||f(T(X)) + f(X) - 2y| \\
&\leq 2W_1 W_0 (W_1 W_0)^\top \tau_T + C|A_T X - b_T - X||f(A_T X + b_T) - f(X) + f(X) + f(X) - 2y| \\
&\leq 2W_1 W_0 (W_1 W_0)^\top \tau_T + C|A_T X - b_T - X|(|f(A_T X + b_T) - f(X)| + |f(X) + f(X) - 2y|) \\
&\leq 2W_1 W_0 (W_1 W_0)^\top \tau_T + C|A_T X - b_T - X|(C|A_T X - b_T - X| + |f(X) + f(X) - 2y|) \\
&\leq 2W_1 W_0 (W_1 W_0)^\top \tau_T + C|A_T X - b_T - X|(C|A_T X - b_T - X| + 2\varepsilon).
\end{aligned}
$$

$\square$

### B.2  PROOF OF PROPOSITION 3

*Proof.* We follow a similar proof strategy as in Proposition 2 where we use Dynkin's formula to compare the solution at different points. The first assumption follows from placing a uniform distribution over the ReLU activation functions being nonzero. Note that from the intermediate value theorem there exists $\epsilon^\star \in (0, 1)$ such that $q(\epsilon^\star) = 1 - \epsilon^\star$. From Dynkin's formula, we can compare the expected loss at the two points by

$$u(X_\vee, y_\vee) = (f(X_\vee) - y_\vee)^2 + \mathbb{E}\left[\int_0^{\tau_{y_\vee}} \nabla^2 (f(X_s) - y_s)^2 \mathrm{d}s \mid X_0 = X_\vee, y_0 = y_\vee\right]$$

and for the underrepresented class,

$$u(X_\wedge, y_\wedge) = (f(X_\wedge) - y_\wedge)^2 + \mathbb{E}\left[\int_0^{\tau_{y_\wedge}} \nabla^2 (f(X_s) - y_s)^2 \mathrm{d}s \mid X_0 = X_\wedge, y_0 = y_\wedge\right].$$

Since $2\epsilon^\star W_1 W_0 (W_1 W_0)^\top \leq \nabla^2 (f(X_s) - y_s)^2 \leq 2W_1 W_0 (W_1 W_0)^\top$, we let the lower bound correspond to the integrand of the solution at $(X_\wedge, y_\wedge)$ and the upper bound correspond to the solution at $(X_\vee, y_\vee)$. This gives us with probability $1 - \epsilon^\star$,

$$(f(X_\wedge) - y_\wedge)^2 + 2\epsilon^\star W_1 W_0 (W_1 W_0)^\top \tau_{y_\wedge} \leq u(X_\wedge, y_\wedge) \tag{8}$$

and with probability 1:

$$u(X_\vee, y_\vee) \leq (f(X_\vee) - y_\vee)^2 + 2W_1 W_0 (W_1 W_0)^\top \tau_{y_\vee}. \tag{9}$$

Setting $\tau_{y_\wedge} \geq \frac{\tau_{y_\vee}}{\epsilon^\star}$, the left hand side of equation 8 is greater than the right hand side of equation 9. This achieves the desired result. $\square$

### B.3 PROOF OF APPROXIMATION

We now illustrate how the approximation relates to the expected loss landscape through the following lemma:

**Lemma 1** (Approximation Error). *Let $\ell_{f_\theta}(X)$ represent the loss and $\bar{\ell}_{f_\theta}(X)$ represent the expected loss over the domain. Then, under the Brownian bridge loss approximation,*

$$\nabla^2 \bar{\ell}_{f_\theta}(X) - \nabla^2 \ell_{f_\theta}(X) = \mathbb{E}_{\mathrm{BB}}\left[\int_0^\tau \ell_{f_\theta}(X_s)\mathrm{d}s\right].$$

*That is, the difference between expected loss and the real is given by the value of the Brownian bridge objective.*

*Proof.* Recall that Dynkin's formula states

$$\underbrace{\mathbb{E}[\ell_{f_\theta}(X_\tau) \mid X_0 = x]}_{\bar{\ell}_{f_\theta}(x)} = \ell_{f_\theta}(x) + \underbrace{\mathbb{E}\left[\int_0^\tau \nabla^2 \ell_{f_\theta}(X_s)\mathrm{d}s \mid X_0 = x\right]}_{\mathrm{BB}}$$

where the expectation is taken over Brownian motion sample paths $X_t$. The left hand side is equal to the expected loss landscape $\bar{\ell}$, the Brownian bridge loss minimizes the BB term.

$\square$

## C  LOWER ORDER DERIVATIVES

As mentioned in the main text, we also have control over the parameters in the low order derivatives through an importance sampling technique. Here we will review how this is used to modify the Monte Carlo scheme such that certain points are given more weight in the loss. For additional details, please see Øksendal (2003) and Pardoux and Rașcanu (2014). For convenience, we will consider the stochastic process $z_t \equiv (x_t, y_t)$.

### C.1  EXPONENTIAL MARTINGALE

Consider first the problem of the Laplace equation solved over a domain $\mathcal{D} \subset \mathbb{R}^{d+k}$ and its connection to Brownian motion. Specifically, we write the solution of

$$\frac{1}{2}\nabla^2 u = 0, \quad z \in \mathcal{D}$$
$$u = \ell, \quad z \in \partial\mathcal{D}$$

according to the following expectation

$$u(z) := \mathbb{E}\left[\ell(z_{\tau_{\partial\mathcal{D}}}) \mid z_0 = z\right]$$

where

$$\tau_{\partial\mathcal{D}} = \inf\{t > 0 \mid z_t \in \partial\mathcal{D}\}$$

and $\mathrm{d}Z_t = \mathrm{d}W_t$.

Now suppose we wish to write the solution for the case where there exist first order derivatives with coefficients $\mu(z) : \mathcal{D} \to \mathbb{R}^{d+k}$ corresponding to

$$\frac{1}{2}\nabla^2 u_\mu + \mu^\top(z)\nabla u_\mu = 0, \quad z \in \mathcal{D} \tag{10}$$
$$u_\mu = \ell, \quad z \in \partial\mathcal{D}$$

which involves the following expectation

$$u_\mu(z) := \mathbb{E}\left[\ell(Z_{\tau_{\partial\mathcal{D}}}) \mid Z_0 = Z\right]$$

with $\tau_{\partial\mathcal{D}}$ defined as before and

$$\mathrm{d}Z_t = \mu(Z_t)\mathrm{d}t + \mathrm{d}W_t. \tag{11}$$

Instead of sampling paths of equation 11, we can instead compute the expectation with the original Brownian motions paths weighted by the *exponential martingale* given by

$$\frac{\mathrm{d}P_\mu}{\mathrm{d}Q} := \exp\left(\int_0^{\tau_{\partial\mathcal{D}}} \mu^\top(Z_s)\mathrm{d}Z_s - \frac{1}{2}\int_0^{\tau_{\partial\mathcal{D}}} \mu^\top\mu(Z_s)\mathrm{d}s\right)$$

where $P_\mu$ denotes the measure associated with the sample paths of equation 11 and $Q$ is the Wiener measure. This then gives us the solution to the new PDE in equation 10 as

$$u_\mu := \mathbb{E}_Q\left[\ell(Z_{\tau_{\partial\mathcal{D}}})\frac{\mathrm{d}P_\mu}{\mathrm{d}Q}\,\middle|\, Z_0 = Z\right]$$

Since we want to minimize $u_\mu$ for boundary conditions dependent on our learned mapping $f$, we can apply Jensen's inequality to get rid of the $\exp$ term in the exponential martingale which may be subject to numerical errors when approximated using an Euler scheme.

## C.2 Choosing the right $\mu$

In our experiments, we choose $\mu$ to be $\xi\nabla\ell(f(x), y)$ for some $\xi > 0$ to get the following PDE

$$\frac{1}{2}\nabla^2 u_\mu + \xi\nabla_z\ell(z)^\top(z)\nabla u_\mu = 0, \quad z \in \mathcal{D}.$$

In practice, one can always make $\mu$ some function (e.g. a neural network) and separately optimize it achieve certain properties of the loss landscape. We consider the gradient of $\ell$ due to the connection between $\ell$ and the uncertainty around a new point. Points with large uncertainty are then given a large weight based on the magnitude of the gradient and influence the loss more.

## D Review of the Feynman-Kac Formula

The Feynman-Kac formula provides a correspondance between expectations of SDEs and PDEs. For full details, we refer to Øksendal (2003, Chapter 9) which describes the correspondence in the case of boundary value problems as used in this work. For self-containment, we provide a review of the formula here. Often, the Fenyman-Kac formula is presented in its form for solving parabolic PDEs. Let $X_t$ satisfy the following Itô diffusion: $\mathrm{d}X_t = \mu(t, X_t)\mathrm{d}t + \sigma(t, X_t)\mathrm{d}W_t$ and consider the following PDE:

$$\frac{\partial u}{\partial t}(t, x) = \nabla u(t, x) \cdot \mu(t, x) + \frac{1}{2}\mathrm{Tr}(\sigma\sigma^\top(t, x)(\mathrm{Hess}_x u)(t, x)) - r(t, x)u(t, x) \qquad (12)$$

with terminal condition $u(x, T) = g(x)$. Then the solution to equation 12 can be represented by the following expectation:

$$u(t, x) = \mathbb{E}\left[g(X_t)\exp\left(-\int_0^t r(X_s)\mathrm{d}s\right)\,\middle|\, X_0 = x\right]. \qquad (13)$$

In the elliptic case, consider the following PDE solved over a domain $\mathcal{D}$:

$$0 = \nabla u(x) \cdot \mu(x) + \frac{1}{2}\mathrm{Tr}(\sigma\sigma^\top(x)(\mathrm{Hess}_x u)(x)) - r(x)u(x), \quad x \in \mathcal{D}. \qquad (14)$$

with boundary condition

$$u(x) = g(x) \quad x \in \partial\mathcal{D}.$$

Then the solution to equation 14 can be represented by the following expectation:

$$u(x) = \mathbb{E}\left[g(X_\tau)\exp\left(-\int_0^\tau r(X_s)\mathrm{d}s\right)\,\middle|\, X_0 = x\right] \qquad (15)$$

where $\tau = \inf\{s > 0 \mid X_s \in \partial\mathcal{D}\}$.

# E    VARIATIONS OF THE BRIDGE COMPARED TO THE ORIGINAL PDE

Here we describe a few more interpretations of the Brownian bridge augmentation as compared to the original PDE that we are solving. Recall that solving the full elliptic PDE by computing hitting times of Brownian motion is prohibitively expensive. We therefore need an algorithm that is more scalable for the typical problems in machine learning. We show now how the bridge can be interpreted through two different frameworks. To do this, we will rely on the following lemma which states that the hitting location of Brownian motion is most likely to be the location closest to the starting point for any stopping time.

**Lemma 2** (Hitting Location of Brownian Motion). *Let $\{(X^{(i)}, y^{(I)})\}_{i=1}^K$ be a set of boundary centroids ordered such that $\|(X^{(i)}, y^{(i)} - (X_0, y_0)\| < \|(X^{(i+1)}, y^{(i+1)} - (X_0, y_0)\|$ for all $i = 1 \ldots K - 1$ and let $\mathcal{B}^{(i)} = \{X, y \| \|X^{(i)} - X + y^{(i)} - y\| \leq \epsilon\}$ for some $\epsilon > 0$. Then,*

$$P(X_\tau \in \mathcal{B}^{(i)}) > P(X_\tau \in \mathcal{B}^{(i+1)})$$

*for all $i = 1 \ldots K - 1$.*

*Proof.* Since $X_t \sim \mathcal{N}(X_0, y_0, t)$, the variance of $X_t$ increases with time. Then for any $\tau > 0$, $P(X_\tau \in \mathcal{B}^{(i)}) \propto \exp(-\|X^{(i)} - X_0 + y^{(i)} - y_0\|/2\tau)$. Since $\int_0^\epsilon \|X^{(i)} - X_0 + y^{(i)} + \varepsilon - y_0\| \mathrm{d}\varepsilon < \int_0^\epsilon \|X^{(i+1)} - X_0 + y^{(i+1)} - y_0 + \varepsilon\| \mathrm{d}\varepsilon$, the probabilities of being in $\mathcal{B}^{(i)} > \mathcal{B}^{(i+1)}$. □

Lemma 2 allows us to sample Brownian bridges with endpoints mapped to the nearest starting points rather than sample paths with random stopping times. Next we describe alternative implementations of the bridge algorithm in terms of Dynkin's theorem and an elliptic PDE with a source term.

Next, we use a fixed $\tau$ for our implementation so we have a fixed time and thus a finite time algorithm for sampling paths for the expectation. Since the variance of the sample paths is given by $\sigma$, we note that there is an ambiguity between $\sigma$ and $\tau$. This is easy to show for two Brownian motions $W_t, B_t$ with $W_{\tau_1} \sim \mathcal{N}(0, \tau_1)$ and $B_{\tau_2} \sim \mathcal{N}(0, \sigma\tau_2)$, then their distributions will be equal if $\sigma = \frac{\tau_1}{\tau_2}$. We then fix $\tau$ in our experiments and optimize over $\sigma$ in the implementation.

## E.1    INTEPRETATION VIA DYNKIN'S FORMULA

Dynkin's formula states the following:

$$\mathbb{E}[\ell(f(X_\tau), y_\tau) \mid X_0 = x] := \ell(f(X_0), y_0) + \mathbb{E}\left[\int_0^\tau \mathcal{A}\ell(f(X_s), y_s)\mathrm{d}s \mid X_0 = x\right]$$

which allows to rewrite the expectation of a stopping time with respect to the integral of the generator $\mathcal{A}$ of $X_s, y_s$ applied to the function $\ell$. In our case, $\mathcal{A}$ is related to the PDE in equation 1 we are trying to impose. We can consider an approximation of this operator through the following

$$\mathbb{E}[\ell(f(X_\tau), y_\tau) \mid X_0 = x] = \mathbb{E}\left[\int_0^\tau \lim_{\Delta t \to 0} \frac{1}{\Delta t}(\mathbb{E}[\ell(f(X_{s+\Delta t}), y_{s+\Delta t})] - \ell(f(X_s), y_s))\mathrm{d}s\right] + \ell(f(X_0), y_0)$$

$$\approx \mathbb{E}\left[\sum_{i=1}^{N_T} \frac{1}{\Delta t}(\mathbb{E}[\ell(f(X_{i+1}), y_{i+1})] - \ell(f(X_i), y_i))\right] + \ell(f(X_0), y_0).$$

These expectations can be computed over sample paths with a fixed $\tau$. In general, $\tau$ can be sampled as some function of the distance between the endpoints.

## E.2    INTERPRETATION VIA AN ELLIPTIC PDE WITH A SOURCE TERM

In the original formulation, all sources of variation arose from the boundary condition which was based on the loss value for a ball around each data point. We can instead think of extending the boundary to be at infinity and only consider instead source terms given by the loss function at different points. This corresponds to the following PDE:

$$\nabla^2 u(X, y) = \ell(f_\theta(X), y), \quad (X, y) \in \mathcal{D} \tag{16}$$

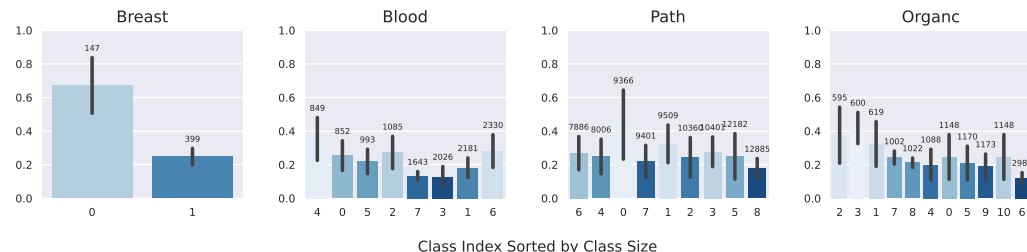

Figure 4: Test data uncertainty score (min-max normalized) of each class for different datasets. X-axis are class indexes sorted by data size; the errorbars present $\pm$ standard deviation; the number above each bar is the class size.

with $u(X, y) = \ell(f_\theta(X), y)$ for $(X, y) \in \partial\mathcal{D}$. This corresponds to the stochastic solution given by

$$u(X, y) := \mathbb{E}\left[\ell(f_\theta(X_\tau), y_\tau) + \int_0^\tau -\ell(f_\theta(X_s), y_s)\mathrm{d}s \mid X_0 = X, y_0 = y\right].$$

Using this representation, we can again sample Brownian bridges between endpoints but instead include an integral over the bridge points instead of computing the expectation over all points along the bridge path. For the new PDE in equation 16, $u$ is no longer harmonic; rather it is subharmonic. Therefore, only the maximum principle is satisfied, not the minimum principle. This is formalized in the following proposition:

**Proposition 4.** *Using the interpretation in equation 16, the expected loss is bounded from above by the loss on the boundary of the domain.*

*Proof.* In equation 16, $u$ is subharmonic and the maximum principle applies. $\qquad\square$

### E.3  MINIMIZING THE SOURCE TERM

Putting the two together, we can minimize the source term to develop a computationally effective way to satisfy the PDE. We make the substitution in equation 16 such that we can guarantee the maximum principle holds. We then apply Dynkin's formula with $\mathcal{A}(\ell(f(X_s), y_s)) = \ell(f(X_s), y_s)$ to get

$$\min \mathbb{E}[\ell(f(X_\tau), y_\tau) \mid X_0 = x] = \min \ell(f(X_0), y_0) + \mathbb{E}\left[\int_0^\tau \ell(f(X_s), y_s)\mathrm{d}s \mid X_0 = x\right].$$

Minimizing this as described in the main text has the advantage of being computationally efficient as well as at the minimum corresponding to the original problem. We can also see how the harmonic property is being violated by checking the value of $\mathbb{E}\left[\int_0^\tau \ell(f(X_s), y_s)\mathrm{d}s \mid X_0 = x\right]$. At the minimum, $\ell(f(X_s), y_s) = 0$ and we will recover the original operator given by $\mathcal{A}\ell(f(X_s), y_) = 0$.

## F  ADDITIONAL EXPERIMENTS

In this section we present additional experiments including ablation studies on some parameters of elliptic training, further empirical results on the robustness of the model trained with elliptic regularization, as well as additional benchmarks for the data imbalance/distribution shift regime against group-informed algorithms.

### F.1  ABLATION STUDY

We provide ablation studies on some hyperparameters used in elliptic training as well as the robust performance against artificial noise induced in test data. For the ablation studies, we mainly focus on the Waterbirds dataset.

The hyperparameters we focus on include $\xi$ and $\sigma_b$. Parameter $\xi$ is related to the strength of the importance weighting, or also as the magnitude of the drift derived from Girsanov's Theorem; $\sigma_b$ is

Table 7: Ablation Study on Waterbirds

|  | Avg(%) | Worst(%) |
| --- | --- | --- |
| Baseline ($\xi = 1, \sigma = 0.1$) | $\mathbf{92.0}_{\pm 0.25}$ | $\mathbf{84.1}_{\pm 1.1}$ |
| No Drift ($\xi = 0, \sigma = 0.1$) | $91.2_{\pm 0.5}$ | $81.7_{\pm 2.6}$ |
| Moderate Drift ($\xi = 2, \sigma = 0.1$) | $92.0_{\pm 0.05}$ | $81.2_{\pm 2.9}$ |
| Large Drift ($\xi = 10, \sigma = 0.1$) | $91.2_{\pm 0.6}$ | $82.5_{\pm 0.6}$ |
| Large Diffusion ($\xi = 1, \sigma = 1$) | $89.9_{\pm 1.1}$ | $75.6_{\pm 0.5}$ |
| Small Diffusion ($\xi = 1, \sigma = 0.01$) | $91.1_{\pm 0.6}$ | $81.1_{\pm 1.8}$ |

Table 8: Noisy evaluation on Waterbirds.

|  | Avg(%) | Worst(%) |
| --- | --- | --- |
| Baseline | $92.0_{\pm 0.25}$ | $84.1_{\pm 1.1}$ |
| Noise Std = 0.1 | $92.1_{\pm 1.6}$ | $79.8_{\pm 3.0}$ |
| Noise Std = 0.5 | $86.0_{\pm 0.8}$ | $71.7_{\pm 4.7}$ |

the diffusion coefficient for data during sampling of Brownian bridges. We show in Table 7 that the importance weighting is necessary to improve model performance in group-imbalanced classification setting, and a moderate diffusion coefficient should be selected to achieve the best possible result.

We also show that models trained with elliptic regularization are robust against noise. We first train a model using elliptic regularization and clean data and then we add artificial Gaussian noise with different standard deviations to the test data. Table 8 showcases the robust result of average and worst-group accuracy.

## F.2 CONTROLLED COMPUTATION EXPENSE

In this section we reduce the number of epoch for Elliptic training such that the number of function evaluation is similar to the benchmarks. Specifically we rerun the MedMNIST experiments: both ERM and mixup is trained with 500 epochs, while Elliptic training is experimented with 5 bridge timesteps and 100 epochs. We show in Table 9 that Elliptic training continues to outperform ERM and mixup with similar number of function evaluations.

## F.3 COMPUTING PAIRWISE DISTANCE

In the main text, we noted that the pairwise distance computation does not affect the model performance. Using the medMNIST datasets, we empirically demonstratet this in Table 10 where the model's robust performance is not affected. We suspect that the randomness in the batch leads to a general diffusive behavior that enforces the elliptic operator.

Table 9: Average and worst class classification accuracy for selected Med-MNIST dataset with a controlled number of function evaluations.

| | Breast | | Blood | | Path | | OrganC | |
| --- | --- | --- | --- | --- | --- | --- | --- | --- |
| Algorithm | Avg(%) | Worst(%) | Avg(%) | Worst(%) | Avg(%) | Worst(%) | Avg(%) | Worst(%) |
| ERM | $80.0_{\pm 3.4}$ | $32.4_{\pm 13.8}$ | $81.4_{\pm 2.7}$ | $56.3_{\pm 17.3}$ | $55.7_{\pm 3.8}$ | $10.7_{\pm 8.6}$ | $80.7_{\pm 2.9}$ | $52.9_{\pm 6.9}$ |
| mixup | $83.1_{\pm 2.4}$ | $47.1_{\pm 10.9}$ | $78.8_{\pm 3.1}$ | $41.7_{\pm 19.3}$ | $54.3_{\pm 2.1}$ | $1.2_{\pm 2.1}$ | $81.3_{\pm 2.2}$ | $56.8_{\pm 11.0}$ |
| Elliptic | $86.7_{\pm 0.9}$ | $67.6_{\pm 4.9}$ | $81.5_{\pm 0.7}$ | $63.4_{\pm 7.0}$ | $57.8_{\pm 0.7}$ | $12.8_{\pm 7.1}$ | $85.8_{\pm 0.6}$ | $64.6_{\pm 6.4}$ |
| E + IW | $86.9_{\pm 1.6}$ | $67.6_{\pm 3.8}$ | $81.5_{\pm 1.0}$ | $62.9_{\pm 9.7}$ | $56.3_{\pm 0.3}$ | $7.1_{\pm 3.5}$ | $86.0_{\pm 1.0}$ | $60.4_{\pm 9.2}$ |

Table 10: Average and worst class classification accuracy for selected Med-MNIST dataset with pairwise distance computation (P-elliptic) and without (elliptic).

| | Breast | | Blood | | Path | | OrganC | |
|---|---|---|---|---|---|---|---|---|
| Algorithm | Avg(%) | Worst(%) | Avg(%) | Worst(%) | Avg(%) | Worst(%) | Avg(%) | Worst(%) |
| Elliptic | $87.6_{\pm1.8}$ | $67.6_{\pm4.4}$ | $85.5_{\pm1.6}$ | $60.2_{\pm14.9}$ | $62.3_{\pm2.5}$ | $26.4_{\pm10.2}$ | $87.7_{\pm0.9}$ | $65.9_{\pm3.4}$ |
| E +IW | $87.3_{\pm0.9}$ | $67.6_{\pm3.0}$ | $84.1_{\pm4.0}$ | $68.0_{\pm13.7}$ | $62.7_{\pm1.8}$ | $20.9_{\pm6.8}$ | $88.0_{\pm0.7}$ | $66.2_{\pm4.8}$ |
| P-Elliptic | $87.3_{\pm1.8}$ | $68.1_{\pm3.9}$ | $85.3_{\pm1.8}$ | $61.9_{\pm13.2}$ | $62.7_{\pm2.3}$ | $24.7_{\pm9.7}$ | $87.9_{\pm1.0}$ | $65.6_{\pm3.3}$ |
| P + IW | $87.6_{\pm1.1}$ | $68.1_{\pm3.2}$ | $84.3_{\pm3.3}$ | $67.0_{\pm10.9}$ | $62.7_{\pm1.4}$ | $23.6_{\pm8.4}$ | $88.2_{\pm0.8}$ | $65.5_{\pm4.5}$ |

Table 11: Robust classification accuracy under imbalance subpopulations and domain shift (grouped-informed).

| Algorithm | WaterBirds | | CelebA | | Camelyon17 |
|---|---|---|---|---|---|
| | Avg(%) | Worst(%) | Avg(%) | Worst(%) | Avg(%) |
| IRM (Arjovsky et al., 2019) | $87.5_{\pm0.7}$ | $75.6_{\pm3.1}$ | $94.0_{\pm0.4}$ | $77.8_{\pm3.9}$ | $64.2_{\pm8.1}$ |
| IB-IRM (Ahuja et al., 2021) | $88.5_{\pm0.6}$ | $76.5_{\pm1.2}$ | $93.6_{\pm0.3}$ | $85.0_{\pm1.8}$ | $68.9_{\pm6.1}$ |
| V-Rex (Krueger et al., 2021) | $88.0_{\pm1.0}$ | $73.6_{\pm0.2}$ | $92.2_{\pm0.1}$ | $86.7_{\pm1.0}$ | $71.5_{\pm8.3}$ |
| CORAL (Sun and Saenko, 2016) | $90.3_{\pm1.1}$ | $79.8_{\pm1.8}$ | $93.8_{\pm0.3}$ | $76.9_{\pm3.6}$ | $59.5_{\pm7.7}$ |
| GroupDRO (Sagawa et al., 2019) | $91.8_{\pm0.3}$ | $90.6_{\pm1.1}$ | $92.1_{\pm0.4}$ | $87.2_{\pm1.6}$ | $68.4_{\pm7.3}$ |
| DomainMix (Xu et al., 2020) | $76.4_{\pm0.3}$ | $53.0_{\pm1.3}$ | $93.4_{\pm0.1}$ | $65.6_{\pm1.7}$ | $69.7_{\pm5.5}$ |
| Fish (Shi et al., 2021) | $85.6_{\pm0.4}$ | $64.0_{\pm0.3}$ | $93.1_{\pm0.3}$ | $61.2_{\pm2.5}$ | $74.7_{\pm7.1}$ |
| LISA (Yao et al., 2022b) | $91.8_{\pm0.3}$ | $89.2_{\pm0.6}$ | $92.4_{\pm0.4}$ | $89.3_{\pm1.1}$ | $77.1_{\pm6.5}$ |
| Elliptic + IW | $92.0_{\pm0.25}$ | $84.1_{\pm1.1}$ | $91.3_{\pm0.3}$ | $77.4_{\pm4.5}$ | $77.9_{\pm3.0}$ |

## F.4 ADDITIONAL COMPARISONS

In this subsection we compare Elliptic regularization training with importance weighting with algorithms that leverage the group-labels during training. This set of algorithms includes IRM (Arjovsky et al., 2019), IB-IRM (Ahuja et al., 2021), V-Rex (Krueger et al., 2021), CORAL (Sun and Saenko, 2016), GroupDRO (Sagawa et al., 2019), DomainMix (Xu et al., 2020), Fish (Shi et al., 2021), LISA (Yao et al., 2022b). We show in Table 11 that our one-stage, group-oblivious training algorithm is comparable to these group-informed algorithms in all three classification datasets. Elliptic regularization achieved the best average and third-best worst group accuracy for waterbirds, results comparable to IRM and CORAL in CelebA, and the best accuracy for Camelyon17.

## G DATASET DETAILS

We briefly describe the contents and goals of each dataset used in our experiments, provide details about data preprocessing, and reference to related publically available codebases.

### G.1 BALANCED/IN-DISTRIBUTION EXPERIMENTS

**CIFAR10, CIFAR100, Tiny-Imagenet 200:** For these well-known, small-scale image classification datasets, we followed preprocessing pipeline in Zou et al. (2023), which is publically available in the mixupE repository.

**AirFoil, NO2:** These are publically available tabular datasets with continuous labels for regression tasks. We applied preprocessing to these two datasets following Yao et al. (2022a). The preprocessing is publically available in the c-mixup repository

The AirFoil (AirFoil Self-Noise) dataset (Brooks et al., 2014) aims to predict acoustic testing results given the physical features of two and three-dimensional airfoil and the wind tunnel environment.

There are 1503 instances in AirFoil; a min-max normalization is applied to the data; there are 1003, 300, and 200 data instances for training, validation, and testing respectively.

The NO2 dataset is included in Statlib (Kooperberg, 1997). The dataset studies air pollution and its relationship to traffic volume and meteorological variables. The dataset is collected by the Norwegian Public Roads Administration on Alnabru in Oslo between October 2001 and August 2003. The response variable column 1 consists of the hourly logged concentration of NO2 particles. There is no preprocessing applied; we split 200, 200, and 100 for training, validation, and testing respectively.

### G.2 GROUP IMABLANCE/DISTRIBUTIONAL ROBUST EXPERIMENTS

### G.2.1 CLASSIFICATION

**Waterbirds, CelebA, and Camelyon:** For these three datasets, we followed the preprocessing of Han et al. (2022). We applied the same data preprocessing, which is available in the UMIX repository.

The Waterbirds dataset (Koh et al., 2021) aims to classify whether the bird is a waterbird or a landbird. This dataset has four predefined subpopulations including *landbirds on land*, *landbirds on water*, *waterbirds on land*, and *waterbirds on water*. Imbalanced group/subpopulation exist in the training set: the largest subpopulation is *landbirds on land* with 3,498 samples, while the smallest subpopulation is *landbirds on water* with only 56 samples.

CelebA (Sagawa et al., 2019) is a well-known large-scale face dataset. We predict the color of the human hair as *blond* or *not blond*. There are four imbalanced subpopulations based on gender and hair color including *dark hair, female*, *dark hair, male*, *blond hair, female* and *blond hair, male* with 71,629, 66,874, 22,880, and 1,387 training samples respectively.

Camelyon17 (Bandi et al., 2018) is a pathological image dataset with over 450, 000 lymph node scans for predicting the existence of cancer tissue in a patch. The training data consists of scans from three hospitals, while the validation and test data are sampled from other hospitals. Distribution and domain shifts exist in this data as the classification requires generalization across different hospitals and coloring methods. Due to the complexity of the data, especially considering that different coloring methods are observed even in samples from the same hospital, there are no reliable subpopulation labels of Camelyon17. We applied the official split scheme of this dataset.

### G.2.2 REGRESSION

Crime consists of demographic and economic statistics of different communities which For SkillCraft, RCF-MNIST is a simulated dataset derived from Fashion-MNIST (Yao et al., 2022a).

**SkillCraft, Crime:** Both of these datasets are obtained from the UCI data repository. The preprocessing applied in our experiments is also publically available in the c-mixup repository

SkillCraft (SkillCraft1 Master)(Blair et al., 2013) contains video game telemetry data from real-time strategy (RTS) games to explore the development of expertise. The goal is to predict input action latency based on 17 player-related parameters in the game, such as the Cognition-Action-cycle variables and the Hotkey Usage variables. Missing data are filled by mean padding on each attribute. Levels of competitors, identified through "League Index" variable, are used as the domain information. We split 4,1,3 domains into training, validation, and testing subsets, which contain 1878, 806, 711 data instances, respectively. We train to predict the mean action latency of video game players in perception-action cycles; we treat "LeagueIndex" as domain information and generalize predictions across different leagues.

Crime (Communities And Crimes)(Redmond, 2009) is a tabular dataset that aims to predict violent crimes per capita. The 122 attributes of this dataset combine socio-economic data from the 1990 US Census, law enforcement data from the 1990 US LEMAS survey, and crime data from the 1995 FBI UCR. Following the description of Yao et al. (2022a), we applied a min-max to normalize all numeric features into [0,1]. The missing values are filled with the average values of the corresponding attributes. The 46 different State-IDs are used as the domain information, and we split the dataset into training, validation, and test sets into subsets containing 31, 6, and 9 disjoint domains. There are 1,390, 231, and 373 instances in the training, validation, and testing subsets respectively. We train to

predict the total number of violent crimes(per 100K population) of some states and aim to generalize to unseen states.

**RCF-MNIST** The RCF-MNIST (Rotated-Colored Fashion-MNIST) is a simulated dataset based on Fashion-MNIST Yao et al. (2022a). The main goal of this simulation is to understand the model performance under spurious correlation in the training dataset. The simulation for the training set contains two steps: for a specific data sample, 1) choose a normalized angle of rotation $g \in [0, 1]$, this angle of rotation is used as the label; then 2) color the normalized RGB vector of the original white pixel into $[1 - g, 0, g]$. The color and the angle create a spurious correlation. For testing data, the coloring is reversed; hence the testing data would have the coloring of $[g, 0, 1 - g]$ for a chosen angle of rotation $g$. We used an 80-20 split to split the training and testing data. We train the model to predict the rotation angle of the object in the images.

### G.3 Class Imbalance MedMnist Experiments

The Med-MNIST dataset consists of medical image datasets of different scales. The details of these datasets can be found in Yang et al. (2023a). We are using the official training and testing split of these datasets. All of the images for this set of experiments are normalized to mean = 0.5 and std = 0.5. Imbalanced class exists in all of the selected datasets in our experiments including BreastMNIST, BloodMNIST, PathMNIST, and OrganCMNIST.

BreastMNIST aims to use ultrasound images to identify the existence of malignant breast cancer. There are 780 images in total where a 70-10-20 data split for training, validation, and testing is applied. Among the training data, there are 147 malignant labels and 399. An official data split of 70-10-20 is applied for each dataset to construct the training, validation, and testing subsets respectively.

BloodMNIST consists of microscopic images of blood cells. The 17092 images are obtained from individuals without infection, hematologic, or oncologic disease and free of any pharmacologic treatment during blood collection. The goal is to identify the 8 different cell types that exist in the dataset. An official data split of 70-10-20 is applied for each dataset to construct the training, validation, and testing subsets respectively.

PathMNIST is collected from 100,000 non-overlapping image patches from hematoxylin & eosin stained histological images. The dataset aims to classify the 9 types of tissues. According to the description of the official data splits, the training and validation data is obtained from one clinical center, while the test data is curated from a different clinical center. Hence, to test performance under class imbalance and avoid additional distribution shifts, we only used the official training split as training data and validation data as testing data in our experiments.

OrganCMNIST is derived from 3D computed tomography (CT) images from Liver Tumor Segmentation Benchmark (LiTS). These 3D images are then preprocessed into different views (Yang et al., 2023a). The OrganC images consist of the coronal view of these 3D images. The goal is to classify the 11 body organs from these CT scans. The official data split is applied for this dataset. There are over 10,000 samples in this dataset.

### G.4 Preprocessing

Our data preprocessing follows the preprocessing pipeline of previous works such as mixupE, UMIX, and c-mixup. For CIFARs, standard normalization is applied; for tiny-imagenet, normalization, RandomCrop (to height=width=64), and RandomHorizontalFlip are applied; for celebA and Camelyon17, normalization and RandomHorizontalFlip is applied; waterbirds' transformation includes normalization, RandomHorizontalFlip and a RandomResizedCrop to crop the resolution to 224*224. The normalizations applied to above image datasets use the recommended means and standard deviation for the respective datasets. Min-max scaling is applied to all features of the regression datasets. We utilize the recommended preprocessing pipeline for MedMNIST datasets, which normalizes the data to 0.5 mean and 0.5 standard deviation.

Table 12: Architecture and Hyperparameter settings for each dataset/experiments. Optim stands for optimizer; Mom stands for momentum; WD stands for weight decay; and LR stands for learning rate. Note, * means the learning rate is annealed by a factor of 10 on epoch 100 and 150.

| Dataset | Epoch | Batch Size | Optim | Mom | WD | LR | $n_b$ | $n_t$ | $\sigma_b$ |
|---|---|---|---|---|---|---|---|---|---|
| CIFAR10/CIFAR100 | 200 | 100 | SGD | 0.9 | $10^{-4}$ | $1^{-1}*$ | 1 | 5 | 0.05 |
| Tiny-ImageNet 200 | 200 | 100 | SGD | 0.9 | $10^{-4}$ | $1^{-1}*$ | 1 | 5 | 0.05 |
| AirFoil | 100 | 16 | Adam | - | - | $1^{-2}$ | 20 | 5 | 0.05 |
| NO2 | 100 | 32 | Adam | - | - | $1^{-2}$ | 10 | 5 | 0.01 |
| Waterbirds | 200 | 32 | Adam | - | - | $1^{-5}$ | 1 | 5 | 0.1 |
| CelebA | 50 | 128 | SGD | 0.9 | 0.1 | $1^{-4}$ | 1 | 5 | 0.1 |
| Camelyon17 | 5 | 32 | SGD | 0.9 | 0.01 | $1^{-5}$ | 1 | 5 | 0.1 |
| Crime | 100 | 16 | Adam | - | - | $1^{-3}$ | 10 | 5 | 0.05 |
| SkillCraft | 100 | 32 | Adam | - | - | $1^{-2}$ | 10 | 5 | 0.05 |
| RCF-MNIST | 30 | 64 | Adam | - | - | $1^{-5}$ | 10 | 5 | 0.05 |
| MedMNIST | 500 | 500 | Adam | - | - | $1^{-3}$ | 1 | 10 | 1 |

## H  HYPERPARAMETERS

Now we describe the hyperparameters/architecture information for each set of experiments. For all of our bridge samples, we sampled according to a uniform discretization over the time range $[0, 1]$, except for MedMNIST which we used time range $[0, 0.01]$. We use $n_b$ to denote the number of bridges, and $n_t$ to determine the number of discretization. We use $\sigma_b$

All related training hyperparameters for all experiments are listed in Table 12. For all experiments, parameter $\xi = 1$

### H.1  ARCHITECTURES FOR BALANCED CLASSIFICATION

We followed mixupE's experiment format (Zou et al., 2023) and used PreActResNet50, PreActRes-Net101, and Wide-ResNet-28 for benchmarking on CIFAR10/CIFAR100. We used PreActResNet18, PreActResNet34, PreActResNet50 for benchmarking on Tiny-imagenet200. These are implementations without pre-trained weights.

### H.2  ARCHITECTURES FOR REGRESSION

We applied a 3-layer neural network with hidden layer dimension=128 and the LeakyReLU activation with negative_slope=0.1, which is the same as c-mixup for comparing the regression result (Yao et al., 2022a).

### H.3  ARCHITECTURES FOR GROUP-IMBALANCE/DISTRIBUTION SHIFT CLASSIFICATION

To be consistent with previous work Han et al. (2022), for waterbirds and CelebA, we trained on Pytorch implementation of ResNet50 that is pre-trained on ImageNet; for Camelyon17, we trained on Pytorch implementation of DenseNet121 without pre-trained weights.

### H.4  ARCHITECTURES FOR REGRESSION WITH DISTRIBUTION SHIFT

For Crime and SkillCraft, we used a 3-layer neural network with hidden layer dimension of 128 and the LeakyReLU activation function with negative_slope parameter of 0.1. For RCF-MNIST, we used ResNet18 pre-trained on ImageNet, but only up to the second to last layer, as a feature extractor. Then we applied a linear layer to predict the angle.

## H.5 Architectures for MedMNIST imbalance classification

For all of the MedMNIST datasets, we used a 2-layer neural network with hidden layer dimension of 512. Batch normalization is applied to the output of the first layer.

