# OpenReview forum: "Elliptic Loss Regularization"
_ICLR.cc/2025/Conference — ICLR 2025 Poster_

### Official Review · Reviewer_rpmH · 2024-10-28

**Soundness:** 2
**Presentation:** 2
**Contribution:** 2
**Rating:** 5
**Confidence:** 2

**Summary:**

This work proposes a regularization technique for a loss landscape where ideas are taken from Partial Differential Equations (PDEs) so that one can impose properties on the landscape. The loss landscape satisfies an elliptic operator allowing thus to bound it on the interior of the domain using the observed points. By doing so one can anticipate the behaviour of the error for points outside the traing set and thus addressing  two important problems : Data shifts and Data imbalance.
In this context, a tractable computational method that approximate the elliptic operator is proposed and various experiments are done trying to show its efficiency.

**Strengths:**

This paper addresses an important problem in machine learning : data shift and data imbalance

**Weaknesses:**

Although the addressed problem is important the proposed technique to tackle it is not clear nor justified.
First the related work is poor and it's not related to what is proposed in the paper. Thus, after some enumeration of past work on mixup, it's just stated that "these findings motivate our work where we consider a different type of regularization and its implication on the loss values". This should be motivated and explained.
Then, ERM vs TRM is introduced by saying that the latter may cover more of D depending on how \phi is chosen, but nothing is said on the price to pay for that.
The equation (1) (line 171) is not explained nor justified and as  this equation is the core of the proposed approach, it would be beneficial to explain its derivation
How the equation (1) leads to equation (3) ? the derivation is not clear and there is no details in the appendix
After that, the generalization of the operator through equations (4) and (5) is not clear and the Appendix does not help for that

**Questions:**

1) How the PDE can be justified here ? Why not using OOD technique (for data imbalance) and domain adaptation (for data shift) ?
2) there is a reference to the Dirichlet problem in the introduction but nothing is said about it in the rest of the paper.
3) Why you did not compare to a recent Mixupe approach as Yingtian Zou (2023) (see the last reference).
4) Experiments reflect Mean and Worst, why just these two metrics and not for instance the Best ?

---

> ### Author Response · Authors · 2024-11-17
>
> Thank you for your constructive feedback and efforts in writing the review.
> We hope that we adequately address all the concerns in the review, please let us know if there is anything else that requires discussion.
>
> ### **Related Work**
>
> The closest augmentation based method to the proposed method is mixup, especially its implication of solving a regularized problem through augmentation as pointed out in the related work. Using TRM to solve a regularization problem motivates us to develop our method of using diffusion to solve an optimization problem regularized by a PDE. We have updated the related work accordingly to better demonstrate our motivation stemming from TRM and mixup.
>
> ### **Theoretical Development**
>
> The motivation for constraining the regularization with equation (1) is its theoretical properties, as stated in the introduction (additionally see our response below). The theoretical development from equation 1 to equation 3 follows the application of Feynman-Kac theorem, then we applied the Girsanov theorem to obtain equations 4 and 5.
>
> ### **PDE Justification**
>
> The PDE is justified for two reasons: 1) the theoretical properties described throughout the paper that can be used for the problems we're interested in tackling and 2) the computationally scalable approach that we use to implement the method.
> As such, we provide theoretical evidence for how the PDE perspective provides a unified approach to addressing the problems of distribution shift and imbalance.
> We further show that the PDE approach provides some empirical improvement when compared to other imbalance/shift approaches as the reviewer mentions.
>
> ### **Dirichlet Problem**
>
> We included a reference to a Dirichlet problem at the beginning to bridge the gap to traditional PDE disciplines.
> In the revision, we removed the reference to a Dirichlet problem to avoid confusion.
>
> ### **Comparison to MixupE**
>
> We did include a comparison to the MixupE approach (please see Tables 1, 2, and 6).
>
> ### **Best Group**
>
> For data imbalance experiments, we are interested in quantifying worst group performance which is usually the one that has the fewest samples as average performance across all groups.
> We want to balance the average performance to the worst case performance.
> Best group performance may not be as meaningful since, for example, a method can overfit to a particular group.
> Finally, these are the metrics commonly reported within the literature for assessing performance for group imbalance (see the references for comparable tables).

---

> > ### Author Response · Authors · 2024-11-22
> >
> > Many thanks for your review and feedback. We think that we responded to all of the concerns raised in the review, and as the response period comes to an end, we wanted to make sure that all comments were addressed and accounted for in the revision. Please let us know if there is anything that we should discuss before the end of the period.

---

> > > ### Author Response · Authors · 2024-11-27
> > >
> > > Thank you again for your review and feedback. We hope that we addressed your concerns specifically on why the PDE justification is needed, the modification of the manuscript to remove references to the Dirichlet problem and the rewriting of the related work, the comparison to MixupE, and how the connection between stochastic processes and PDEs provides the derivation of the sampling strategy. We also encourage the reviewer to take a look at our response to Reviewer `tfwK` where we noted similar training time with MixupE and improved training time and performance over Umix on the robust classification dataset. Please let us know if there is anything else that requires clarification, and we hope that you will consider raising your score.

---

> > ### Comment · Reviewer_rpmH · 2024-11-28
> >
> > Thanks for the time and efforts for answering my questions. Reading other reviews and re-reading the article, I have decided not change my score.

---

### Official Review · Reviewer_CpbM · 2024-10-30

**Soundness:** 3
**Presentation:** 2
**Contribution:** 3
**Rating:** 6
**Confidence:** 3

**Summary:**

The paper proposes a new regularizer based on elliptic loss landscapes. The high-level idea is to imbue the loss landscape of a model with a certain kind of smoothness that allows the loss to be bounded in unseen regions of the landscape. This allows the loss to be bounded under data distribution shifts and data imbalance. In practice, the regularizer works by training on the loss evaluated on paths between data points. Experiments show the experiment is effective on CIFAR, TinyImagenet and distribution shift tests.

**Strengths:**

**Originality**

The paper seems quite original- I don't believe previous works have considered using elliptic loss landscapes and the resulting bounds one can derive (in Proposition 1) seem quite powerful. There are previous works exploring other forms of data augmentation (such as mixup) to improve generalization, but I believe few have the strong theoretical guarantees of the proposed method.

**Quality**

Overall, the quality of the paper is strong: the theory looks sound and experiments seem adequate.

**Clarity**

The main text of the paper is adequately written and figures and tables are well-illustrated.

**Signifiance**

The paper proposes a new way to bound the loss on unseen points far from any previously observed points, which is generally a very challenging problem. As such, I expect the theory proposed in this paper alone to have great significance to researchers finding ways of improving generalization, and I expect future work would build on the theory in this paper.

**Weaknesses:**

Regarding the theory, the practical algorithm requires sampling which would likely lead to a gap between the true elliptic loss and the computed loss. Can the authors comment on whether Proposition 1 would theoretically still hold under the sampling technique? Ideally, the error resulting from the sampling process can be bounded.

Also, unfortunately, it seems that the improvement of the method over previous techniques (particularly mixupE) seem small for CIFAR which raises questions about how effective it is at in-distribution generalization. Similarly, the results for domain shift in Table 4 and Table 5 do not seem conclusive in favoring the proposed regularizer.

Clarity-wise, the introduction of elliptic loss landscapes and the proposed algorithm seem a bit confusing due to the complexity of the technique. I would suggest adding an additional figure in the main paper illustrating the nice properties of the loss landscape described by eqns 1 and 2. I would also suggest including an algorithm in the main paper to accompany section 5.1

**Questions:**

Can the authors bound the error of the sampling procedure?
Can the authors explain why the performance of the method is worse in some-cases than other baselines (such as mixupE)?

---

> ### Author Response · Authors · 2024-11-17
>
> Thank you for all your time and effort in reviewing and providing commentary on the manuscript.
> Below we respond to all of the concerns, please let us know if there's anything else that we should address.
>
> ### **Proposition 1 with Approximation**
>
> This is a good point, the reviewer is correct there is some error associated with sampling with a discrete scheme such as Euler-Maruyama.
> The error with Euler-Maruyama can be reduced by decreasing the step size, we included a reference to Chapter 7.2 of [1] in the revision. This is going to be the main source of error introduced by the numerical scheme.
>
> The other consideration is the difference between Brownian bridges which we sample versus unconstrained sample paths.
> In the case of the Laplace equation, sampling bridges with endpoints according to a distribution based on minimum distance is exact (which we discuss in Lemma 1 in appendix E). While sampling should be according to the pairwise distance, sampling without the pairwise distance would induce a first order term but would not remove any of the theoretical properties of the proposition.
>
> Finally, in terms of training behavior, the minimum bound in proposition 1 could be violated if the loss is very large for points along the bridge.
> This would mean that $\ell(f(x),y)$ is not harmonic but is subharmonic, meaning the minimum principle would not hold, only the maximum principle (i.e. the lower bound does not exist but the upper bound does).
> At the minimum when $\ell(f(x),y) =0$, though, approximation guarantees that proposition 1 holds.
>
> ### **Adding Figure and Algorithm**
>
> We thank the reviewer for these suggestions. Due to page limitations, the proposed algorithm outlined in section 5.1 is presented in Appendix A. We updated Figure 1 to illustrate the properties of a loss landscape that satisfies eq (1).
>
> ### **Error of Sampling Procedure**
>
> The error of the sampling procedure is based on the discretization of the sample paths. We updated the manuscript to include a discussion on this as well as the reference to [1, Theorem 7.15] which describes the optimal convergence rate for these types of equations.
>
> ### **Performance of Different Experiments**
>
> The reviewer brings up an interesting question. We first note that our method is universal but is comparable to the methods built specifically for particular settings. We hypothesize that the experiments with a more complex dataset (i.e a larger number of classes, a larger number of subpopulations and groups) are more effective with the regularization since datasets with more classes/groups have a more complex domain that can be better estimated with elliptic regularizers as the data-augmentation spend more of its time in the interior of the domain.
>
> [1] Graham, Carl, and Denis Talay. Stochastic simulation and Monte Carlo methods: mathematical foundations of stochastic simulation. Vol. 68. Springer Science & Business Media, 2013.

---

> > ### Comment · Reviewer_CpbM · 2024-11-19
> >
> > Thank you for your timely and thorough response.
> >
> > My concerns have been addressed and I remain in favor of acceptance.

---

### Official Review · Reviewer_tfwK · 2024-11-02

**Soundness:** 4
**Presentation:** 4
**Contribution:** 3
**Rating:** 8
**Confidence:** 4

**Summary:**

This paper presents a novel regularizer for machine learning problems. The key insight is to construct a training objective (a “loss landscape”) that solves an elliptic PDE with boundary conditions specified by the unregularized loss’ values at the training points. This ensures that the regularized objective cannot be too large away from the training data: In particular, the loss landscape obeys a maximum principle, which implies that the regularized loss is upper-bounded by the maximum of the unregularized loss at the training points.

As it is impractical to exactly solve a PDE in each training iteration, the authors propose a numerical scheme to approximate the loss landscape. This scheme amounts to sampling pairs of training samples within each minibatch with importance weights inversely proportional to their distance, and then evaluating the training loss along paths connecting these pairs of points. The authors then prove two propositions to illustrate how the properties of the loss landscape can be used to predict the behavior of their regularized loss under affine distribution shift and data imbalance. They finally conduct a range of experiments which show that their regularizer performs comparably to baseline methods on problems without group imbalance or distribution shift, better than one-stage but often worse than two-stage baselines in settings with unbalanced populations and domain shift, and significantly better than baseline methods in class-imbalanced problems.

**Strengths:**

- Elliptic loss regularization is a neat and well-motivated method for obtaining generalization guarantees via a maximum principle.
- I appreciate the authors including examples of how one might use the properties of their loss landscape function to predict the effects of distribution shifts and class imbalances.
- The experiments are thorough and adequately study the performance of elliptic loss regularization in a variety of plausible settings.
- The paper is well-written and generally easy to follow.

**Weaknesses:**

My main critique of elliptic loss regularization is that it results in sometimes marginal performance improvements (such as in Table 5) over simpler and cheaper baselines. The authors could further strengthen the case for their regularizer by showing that using their numerical scheme, the computational cost of elliptic loss regularization is not much greater than that of simple mixup baselines.

**Questions:**

- Why use Brownian bridges to connect pairs of training samples in the numerical procedure detailed in Sec 5.1? Is there any practical advantage to Brownian bridges compared to simply sampling along line segments connecting pairs of training samples (as with mixup)?
- As I understand, the training algorithm essentially amounts to sampling pairs of nearby training points (distant pairs are presumably sampled with negligible probability), and evaluating the training loss along paths connecting these pairs. If this is correct, is it reasonable to view elliptic loss regularization as a variant of mixup where the interpolated points are more likely to remain on the data manifold? This seems like an intuitive perspective on why one might expect this method to perform well on manifold-supported data embedded in high-dimensional space, which is true of much real-world data under the manifold hypothesis.
- How costly is elliptic loss regularization relative to simpler baselines like mixup? The proposed method’s performance improvements over baselines are sometimes marginal (see e.g. Table 5), and I wonder if the additional cost of the numerical scheme to approximate the regularized training objective is worth the marginal improvements in such problems.
- Why are worst-case results not reported for the Camelyon17 dataset in Table 4?

---

> ### Author Response · Authors · 2024-11-17
>
> Thank you for your time and efforts in reading the manuscript and providing helpful feedback.
> We appreciate the encouragement, and we believe we address all the related concerns below.
> Please let us know if there is anything that requires additional clarification.
>
> ### **Why Brownian Bridges**
>
> The reviewer brings up a good point. The construction of the regularization scheme is derived from the PDE using the connection between the PDE and stochastic processes via the Feynman-Kac theorem, which requires sampling of diffusion paths to the boundary condition. Sampling only on the linear interpolation between two data is not sufficient to compute the expectation in Eq (3) since it restricts the samples to only lie on the line between any two samples rather than being absolutely continuous with respect to the Lebesgue measure. This would look like convective behavior rather than diffusive, which would have different properties, but is an interesting question nevertheless.
>
> ### **Connection to mixup**
>
> The reviewer is correct that the proposed method is related to the mixup algorithm in the sense that both our method and mixup apply data transformation to solve a regularized problem.
> Elliptic regularization, however, solves a different regularized problem compared to the mixup algorithm motivated by the viewpoint of the PDE.
>
> The reviewer also raises an interesting point regarding the sampling scheme. The sampling scheme was set up to approximate the PDE in expectation.
> While the distance based pairing may induce some of the properties that the reviewer mentions regarding the manifold supported data, it is likely the case that the overall diffusive dynamics are contributing to the method's performance since including pairwise calculations has a small effect on the performance.
> However, trying this on explicitly manifold supported data (e.g. graphs/solving on irregular domains) is a great avenue for future work to test this idea.
> We updated the manuscript to include this.
>
> ### **Cost of Imposing Regularization**
>
> The most computationally expensive part of our method is computing pairwise distance for every dataset, which is of time-complexity $\mathcal{O}(n^2)$. However, this can be pre-computed and saved as a dictionary to be called for $\mathcal{O}(1)$ speed during training.
>
> As stated on line 370, none of our results apply this sampling technique because the improvements are negligible (see Appendix F). If we do not compute distances, the bridge paths correspond to drifted Brownian motion which induces a first-order term on the operator. This has the same effect as the change of measure described in the importance weighting section.
>
> Therefore, whether computing the pairwise distance beforehand or not computing the pairwise distance at all, will result in an algorithm that has marginal computational overhead compared to the other benchmarks and the additional cost comes from the additional function evaluation controlled by the number of bridges and the number of time samples for each Brownian bridge. We experiment with controlled function evaluation where we train fewer epochs for elliptic regularization compared to other benchmarks. These results are in Appendix F. We also note that our method trains faster than the SOTA baseline UMIX, and in general has the same number of function evaluations as UMIX and JTT.
> Method | Average | Worst Group | Time |
> | - | - | - | - |
> Elliptic |	93.3$_{\pm 0.35}$   |   85.2$_{\pm 1.5}$   |  8345(s) |
> UMIX   |    92.0$_{\pm 0.1}$    |   83.4$_{\pm 2}$     | 11795 (s)(stage 1: 5918, stage 2: 5877 ) |
>
> Finally, we compared epoch-wise training speed between elliptic regularization and mixupE and the results are hardware dependent. On an NVIDIA RTX 6000, mixupE trains 3 minutes and 12 seconds per epoch while elliptic regularization trains 3 minutes and 23 seconds per epoch. On an NVIDIA RTX 3090, mixupE trains 4 minutes and 51 seconds per epoch while elliptic regularization trains 4 minutes and 20 seconds per epoch. These experiments are conducted by training CIFAR10 with PreActResNet101 architecture; the reported training time is averaged across 10 epochs.
> Every hyperparameter is the same as those reported in the appendix.
>
>
> ### **Camelyon Table**
>
> The worst group is not shown for the Camelyon dataset because the groups are undefined in this dataset due to datashifts occuring within each group but no labels are associated with these shifts.
> We include details regarding the unknown subpopulation are provided in Appendix G.2.1.

---

> > ### Comment · Reviewer_tfwK · 2024-11-19
> > **Response to rebuttal**
> >
> > Thank you for your helpful answers to my questions!

---

### Official Review · Reviewer_6cLH · 2024-11-02

**Soundness:** 3
**Presentation:** 1
**Contribution:** 2
**Rating:** 3
**Confidence:** 3

**Summary:**

The paper proposes a new regularisation scheme, elliptic loss regularisation, based on optimising a pseudo loss surface which is the solution a Dirichlet boundary value problem. This regularisation scheme is designed to increase the robustness of models to data shift and imbalance. The authors perform experiments to verify some of their claims which shows that elliptic loss regularisation increase performance over other regularisation schemes like mixup in several settings.

**Strengths:**

* I believe the paper proposes quite an original way to perform loss regularisation by defining and basing the regularisation on a pseudo loss surface. I also think the way they define this loss surface as the solution to a boundary value problem is an interesting perspective.

* I think that the paper covers a wide range of experimental settings, from data imbalance to distribution shift. Additionally, across most of these settings elliptic loss regularisation has good performance compared with the given baselines.

**Weaknesses:**

* I think the greatest weakness of the paper is the lack of clarity and motivation around the actually implemented approach. If this can be fixed I will likely increase my score. I have listed below a few specific points about clarity.
    * The most significant part of this lack of clarity is how they define the boundary of their pseudo loss surface. I can find nowhere where it is precisely defined and from my understanding for a Dirichlet boundary value problem to be solvable and hence for their theorical motivation to be true the boundary needs to satisfy certain conditions. By reading the paper it seems to me that the authors either assume the boundary to be the set of training data points or the edge of the convex hull of the training points. However, if the boundary is just the training points, then it is discontinuous and to the best of my knowledge not really a boundary in the sense used in defining the Dirichlet boundary value problem. While if it is the edge of the convex hull of the training data then you do not generally have the value of the loss for the whole boundary and hence can’t use it.
    * In section 5.1 they correctly point out that their approximation needs to be motivated by showing that it maintains the important properties of the true solution. However, I could not find anywhere a justification of why their approximation is accurate. This justification should be clearly presented in the paper to see how near the actual method is to the theory.
    * There seems to be an overloading of the word ‘loss’, to make the paper clear I think there needs to be a clear distinction between the true loss and the pseudo loss surface created by the method.
    * Another weakness is that a lot of the theory proved about the robustness of elliptic loss regularisation is only about the pseudo loss surface. However, except at the boundary, no connection is made between this loss surface and the true loss. This currently makes a large part of the theory lack practical use and I think a clear theoretical justification for why the pseudo loss surface will model the true one needs to be given. For example, at the moment, I could propose my own loss surface, perhaps a constant function with its value being the mean of the training loss, and prove similar propositions to propositions 1 to 3 while knowing that my loss surface is not similar to the true one at all.

* While the predictive performance of a model using elliptic loss regularisation is shown to be good, I could not find mention of the computational cost of the method. It seems to me that the method could be quite computationally costly compared to other methods and hence I think a mention of the computational cost of elliptic loss regularisation should be given.

* One final weakness is that the baselines compared to in the paper a relatively old and there are perhaps better ones to use. However, I am not sure what they are and hope that my other reviewers can point out relevant newer baselines, if they exist.

**Questions:**

****1.**** In the experiments are standard data augmentations used in combination with elliptic loss regularisation and the baselines?

****2.**** Is there a typo on line 166 as $u$ there is defined as a function with the first argument being $f(X)$ not $X$.

****3.**** Eq 6. is confusing to me, specifically the term inside the second expectation seems to be constant over the second expectation, is this a typo or am I missing something?

---

> ### Author Response · Authors · 2024-11-17
> **Response to review part 1**
>
> Thank you for your time and efforts in providing comments and feedback on the manuscript.
> We believe we address all the concerns mentioned in the review below and updated the manuscript accordingly, please let us know if there's anything else that requires clarification.
>
> ### **Boundary Definition**
> We consider the boundary to be the union of the boundary of the convex hull of the training data (to ensure a compact set) and the balls surrounding each data point as described in section 4.1.
> The boundary can be thought of as a union of disjoint boundaries with each boundary being the boundary of the ball centered at each data point, which is simply defining multiple boundary conditions.
> This is done in a variety of settings, one of the most common applications is in inverse scattering [1].
> For points on the boundary of the convex hull, a value of the $\ell(f_\theta(X_\tau), y_t)$ does exist and can be computed, though these points do not lie within the training data.
> Note that even if we do not include the boundary of the convex-hull, Dirichlet problems can be defined and are studied in the literature as exterior Dirichlet problems [2].
> We rewrote section 4.1 to make this clearer.
>
> [1] Hariharan, S. I. (1982). Inverse scattering for an exterior Dirichlet problem. Quarterly of Applied Mathematics, 40(3), 273-286.
>
> [2] Meyers, N., & Serrin, J. (1960). The exterior Dirichlet problem for second order elliptic partial differential equations. Journal of Mathematics and Mechanics, 513-538.
>
>
> ### **Properties of the Approximation**
>
> In Lemma 1 in Appendix E we describe the relationship between the Brownian bridges and the probable paths for computing first hitting times of the boundary. We then use these paths, since they can be easily sampled within a finite amount of time, to apply Dynkin's formula to satisfy the PDE.
> Recalling Dynkin's formula $\mathbb{E}[\ell(f(x_\tau), y_\tau) \mid x_0,y_0 = x,y] = \ell(f(x_0),y_0) + \mathbb{E} \left [\int_0^\tau \mathcal{A} \ell(f(x_s),y_s) d s \mid x_0,y_0 = x,y \right]$ we compute the expectation over these bridge paths instead of the unconstrained sample paths which have unknown sampling time where $\mathcal{A}$ is the generator of the stochastic process $x_t,y_t$.
> Recall that the operator we're imposing is $\mathcal{A} \ell(f(x),y) = 0$, but we want to avoid computing $\mathcal{A}$ directly each iteration since it is costly.
> Instead, we solve the problem $\mathcal{A} \ell(f(x),y) = \ell(f(x),y)$ which requires integrating $\ell(f(x),y)$ over the sample paths.
> Then, since we're minimizing $\ell(f(x),y) \geq 0$, at the minimum of $\ell(f(x),y) = 0$ we obtain the desired relationship $\mathcal{A} \ell(f(x),y) = 0$.
> Therefore at minimum, the properties all hold and the expected loss landscape is equal to the true loss.
> Note that if the value is greater than zero then $\ell(\cdot)$ is subharmonic meaning the maximum principle will still hold (but the minimum principle will not).
> However, at minimum it is then equal to the harmonic solution.
>
> ### **Overloading the Word "Loss"**
>
> Apologies for this, we opted to use the term loss landscape for the object that we perform our analysis on and describe it as an expected error.
> We are open to any suggestions for improving the terminology.
>
> ### **Connection Between Expected Loss and True Loss**
>
> This follows the response given above.
> The two are equal when $\mathcal{A} \ell(f(x),y) =0 $.
> Using Dynkin's formula again $E[\ell(f(X_\tau), y_\tau) \mid X_0 = x, y_0= y] = \ell(f(x),y) + E[\int_0^\tau \mathcal{A} \ell(f(X_s), y_s) \mathrm{d}s \mid X_0 = x, y_0= y]$ where $\mathcal{A}$ is the generator of the stochastic process.
> We're interested in comparing $\ell(f(x),y) $ to $E[\ell(f(X_\tau), y_\tau) \mid X_0 = x, y_0= y]$.
> The difference is given by $ E[\int_0^\tau \mathcal{A} \ell(f(X_s), y_s) \mathrm{d}s \mid X_0 = x, y_0= y]$, which, when $\ell(f(X_s), y_s)$  satisfies the PDE, is zero.
> Then the difference between the loss landscape and the true loss is then given by how well the function $\ell(f(X_s),y_s)$ satisfies PDE.
> By the substitution described in the previous point, the better trained the model is, the closer its adherence to satisfying the equation.
> We included this note within the revised manuscript.

---

> > ### Author Response · Authors · 2024-11-17
> > **Response to review part 2**
> >
> > ### **Computation Cost**
> >
> > The reviewer is correct that the proposed method may incur higher computation costs. The most computationally expensive part of our method is computing and pairwise distance for every data pair, which is of time-complexity $\mathcal{O}(n^2)$. However, this can be pre-computed and saved as a dictionary and require only $\mathcal{O}(1)$ speed during training.
> >
> > As stated on line 370, none of our results apply the sampling technique that samples inverse-proportionally with the compute distance, because the improvements are negligible (see Appendix F). This is because if we do not compute distances, the bridge paths correspond to drifted Brownian motion which induces a first-order term on the operator. This has the same effect as the change of measure described in the importance weighting section.
> >
> > Therefore, either computing the pairwise distance beforehand or not computing the pairwise distance at all, will result in an algorithm that has marginal computational overhead compared to the other benchmarks. We also note that our method trains faster than the SOTA benchmark UMIX, which is a two-stage method.
> >
> > | Method | Average | Worst Group | Time (seconds) |
> > | - | - | - | - |
> > | Elliptic |  93.3$_{\pm 0.35}$   |   85.2$_{\pm 1.5}$   | 8345 |
> > | UMIX   | 92.0$_{\pm 0.1}$    |   83.4$_{\pm 2}$  | 11795 (stage 1: 5918, stage 2: 5877 ) |
> >
> > ### **Old Baselines**
> >
> > The baselines we consider are all within the last 4 years and we believe are important/relevant papers, but if the reviewer has newer methods that we should consider, we can include discussion on them.
> >
> > ### **Data Augmentation**
> >
> > Our data preprocessing follows the preprocessing pipeline of previous works such as mixupE, UMIX, and c-mixup. Specifically, for tiny-imagenet a random crop (to height=width=64) and RandomHorizontalFlip are applied; for celebA and Camelyon17, a RandomHorizontalFlip is applied; waterbirds' transformation includes RandomHorizontalFlip and a RandomResizedCrop to crop the resolution to $224\times 224$. All image datasets are normalized to recommended mean and variances. Finally, min-max scaling is applied to all features of the regression datasets. We utilize the recommended preprocessing pipeline for MedMNIST datasets. We listed these details in the appendix of the revised manuscript
> >
> > ### **Typo**
> > We thank the reviewer for catching this typo, we fixed it in the revision.
> >
> > ### **Equation 6**
> > Thank you for pointing this out, we fixed the notation within the revision to have $\ell(f(X_s), y_s)$ in the expectation.

---

> > > ### Author Response · Authors · 2024-11-22
> > >
> > > Thank you again for your review and feedback. We believe that we responded to all of the concerns raised in the review, and as the response period comes to an end, we wanted to make sure that all comments were addressed and accounted for in the revision. Please let us know if there is anything that we should discuss before the end of the period.

---

> > > > ### Comment · Reviewer_6cLH · 2024-11-24
> > > >
> > > > Thank you for answering my questions and for the edits to the manuscript, which did clear up some of the confusing parts of the submission. However, while I think the idea of the paper is interesting and has merits, the edits to the paper do not go far enough to fix my concerns about the clarity and motivation of the actually implemented approach. For example, the definition of the boundary is still unclear in the edited paper and still there is not a good justification of the approximation in the main text. In fact, parts of the approximation might be critical to the performance of the approach. For instance, starting the paths at data points is not currently justified by the theory proposed in the paper. However, it does make the method more similar to mixup methods, giving a possible alternative reason to why the method performs well. Given this, I think it would be interesting to see how well the method performs when sampling the start of the paths from all of D.
> > > >
> > > > Due to the outstanding problems I have stated above about the clarity of the paper, which I see reflected in the other reviews as well, I have decided not change my score.

---

> > > > > ### Author Response · Authors · 2024-11-25
> > > > >
> > > > > Thank you for your response and continued discussion.
> > > > >
> > > > > > the definition of the boundary is still unclear in the edited paper
> > > > >
> > > > > We define what the boundary of the domain is within the text on Line 207. Please let us know what is unclear about this definition, and we would be happy to include additional clarification.
> > > > >
> > > > > > there is not a good justification of the approximation in the main text
> > > > >
> > > > > In the response we showed that the two formulations are equivalent, and we updated section 5.1 again to emphasize this further. Sampling along the bridge path does not make a difference in terms of the theoretical properties since the elliptic operator is being applied by virtue of the sampling distribution. We do not specify that that the loss needs to be minimized uniformly over the domain within the theory, so the sampling strategy does not in any way violate the theory. In other words, sampling bridges with both endpoints in the observed data or one end point within the domain and the other on the boundary are both valid strategies of minimizing the loss. We hope this is clear and please let us know what, in your opinion, would be an appropriate justification since we have shown that these are equivalent problems. We would be happy to include additional clarification.
> > > > >
> > > > > Given that the remaining concerns of the reviewer correspond to minor edits to the paper which we have addressed, we politely ask the reviewer to reconsider their score. With regards to the comment that lack of clarity is “reflected in the other reviews as well”, we note that the other reviewers who responded stated that their concerns were addressed.
> > > > >
> > > > > Thank you again for your time and efforts.

---

### Meta-Review · Area_Chair_TV8t · 2024-12-19

**Metareview:**

The submission proposes a novel neural network regularisation method for improving robustness to distribution shifts. The approach is based on constructing a loss that satisfies an elliptic operator. An efficient optimisation scheme is derived using the Feynman-Kac forumla, resulting in a monte carlo interpretation of the PDE. Experiments in several distribution shift setting confirm the utility of the proposed approach.

The reviewers had diverse opinions about this paper There was general agreement that the approach is novel and the experiments are well-conducted. I think the criticisms of reviewers 6cLH and rpmH are valid—the clarity and related work could be improved—but insufficient to warrant rejecting the paper. In particular, I find that the criticism of reviewer 6cLH relating the a lack of clarity in defining the boundary is not well founded, although the description of the method in the paper assumes a lot of prior knowledge so I can understand how some readers might not find this clear. Reviewer rpmH expressed concerns about the related work section. I think this section is quite light on making connections to existing work. E.g., there should be discussion on prior work looking at robustness to distribution shift; this is a very commonly studied topic in machine learning. However, the experimental comparison does actually consider DRO methods, so I think this weakness is not sufficient enough to reject the paper.

In the camera ready copy of the paper I would encourage the authors to discuss in the related work section the distributionally robust optimisation methods they compare with experimentally. I also note that the authors suggest on line 147 that making $f_\theta$ Lipschitz would be an alternative to their approach, but dismiss it as being too difficult; this is an area that has already been investigated [1] and is often used in the adversarial robustness literature [2]. I would encourage the authors to add some discussion of this as well.

[1] Gouk et al. Regularisation of Neural Networks by Enforcing Lipschitz Continuity. In Machine Learning, 2020.

[2] Zuhike & Kudenko. Adversarial Robustness of Neural Networks From the Perspective of Lipschitz Calculus: A Survey. In ACM Computing Surveys, 2024.

**Additional Comments On Reviewer Discussion:**

There was not much discussion from the reviewers, with the exception of reviewer 6cLH. See above for my reasoning on how their view impacted by decision.

---

### Decision · Program_Chairs · 2025-01-22

Accept (Poster)